

# How downstream sub-basins depend on upstream inflows to avoid scarcity: typology and global analysis of transboundary rivers

Hafsa Ahmed Munia[1*], Joseph Guillaume[1], Naho Mirumachi[2], Yoshihide Wada[3,4,5], Matti Kummu[1]

1. Water and Development Research Group, Aalto University, Tietotie 1E, Espoo 02150, Finland

2. Department of Geography, King's College London, Strand, London WC2R 2LS, UK

3. NASA Goddard Institute for Space Studies, 2880 Broadway, New York, NY 10025, USA

4. Center for Climate Systems Research, Columbia University, 2880 Broadway, New York, NY 10025, USA

5. Department of Physical Geography, Faculty of Geosciences, Utrecht University, Heidelberglaan 2, 3584 CS Utrecht, The Netherlands

* *Correspondence to*: Hafsa Ahmed Munia ( hafsa.munia@aalto.fi)

**Abstract:** Countries sharing river basins are often dependent upon water originating outside their boundaries; meaning that without that upstream water, water scarcity may occur, with flow-on implications for water use and management. We develop a formalisation of this concept using water stress and shortage as indicators of water scarcity, and including both persistent and occasional scarcity. Dependency occurs if water from upstream is needed to avoid either persistent or occasional water scarcity. This can be diagnosed by comparing different types of water availability on which a sub-basin relies, starting with reliable local runoff (available even in a dry year), followed by less reliable local water (available in the wet year), reliable dry year inflows from possible upstream area, and finally less reliable wet year inflows from upstream. At the same time, possible upstream water withdrawals reduce available water downstream, influencing the latter two water availabilities. In this paper, we further present a typology describing how scarcity and dependency evolve in transboundary river basins, and use this typology for a global analysis of transboundary river basins at the scale of sub-basin areas (SBAs). Four groups of SBAs are identified that experience scarcity and dependency differently depending on their i) location in the basin, and ii) hydro-climate characteristics, specifically the level of reliable support provided by natural upstream inflows. Each group has its own set of transitions in scarcity and dependency category, driven by changes in local water demand and/or upstream withdrawals. Our results show that almost 932 million people (33% of the total transboundary population) live in SBAs that are dependent on upstream water to avoid stress because of their own water use, while 464 million people (17% of the total transboundary population) live in SBAs dependent on upstream water to avoid possible shortage. The identification of groups and their transitions enables discussion of the pathways SBAs might take in future, potentially contributing to further refined analysis of inter and intrabasin hydro-political power relations and strategic planning of management practices in transboundary basins.

## 1. Introduction

While water is a renewable resource, its availability is still finite. As population and water use grows, water may become scarce. If local precipitation is insufficient to meet needs, a region may draw on external water resources, both physical and virtual (through food and goods trade) (Hoekstra and Chapagain 2011). External water resources constitute a considerable part of the total renewable water of some countries, and create hydrological, social and economic interdependencies between countries (Hoekstra and Mekonnen 2012). Transboundary water resources crossing national borders are a high-profile example. In basins like the Nile and Rio Grande, water availability of the downstream countries (Sudan, Egypt, Mexico) is



related to the upstream precipitation patterns and upstream water use (Drieschova et al. 2008). Transboundary waterbodies cover almost half of the earth's land surface, and are home to about 1/3 of the world's population (UN Water 2013). Increase in water demand is among the main factors that are responsible for water scarcity in most of the transboundary river basins (Degefu et al. 2016). Uncontrolled land and water development in upstream regions can escalate risk of water variability in the downstream region (Al-Faraj and Scholz 2015, Drieschova et al. 2008, Veldkamp et al. 2017). Concerns about water variability are already considered one of the most important issues for international co-operation and conflict concerning shared water basins (Beck et al. 2014).

Thus, 'Hydro-political dependency' in the transboundary river basin becomes an important geopolitical issue affecting the power relation between riparian countries, and potentially sovereignty at national level (Brochmann and Gleditsch 2012, Giordano and Wolf 2003, Gleick 2014, Jägerskog and Zeitoun 2009, Mirumachi 2015, Mirumachi 2013, Wolf 1998, Wolf 1999, Wolf 2007) . It has already been recognized that upstream water use has considerable impact on downstream water scarcity by some regional and global studies (Munia et al. 2016, Nepal et al. 2014, Scott et al. 2003, Veldkamp et al. 2017). When population (or withdrawal) grows, downstream countries eventually become more reliant on the water available from upstream parts of a basin in order to satisfy their needs.

To complement this existing work, in this study we aim to analytically explore the intuitive concept that *upstream water dependency occurs if water from upstream is needed to avoid scarcity*. We argue that a sub-basin therefore experiences a 'hidden' dependency: a downstream part of a basin might be avoiding water scarcity only thanks to upstream inflows, and may not actually realise it until those inflows are no longer available due to increased upstream withdrawals or lower runoff due to potential climate change impacts. Identifying a dependency involves comparing whether scarcity occurs when sub-basin water availability is calculated using either *local runoff*, *natural discharge* (sum of local runoff and upstream runoff), and *actual discharge* (subtracting upstream water withdrawals from natural discharge).

We can classify regions according to their dependency category. Based on the role of upstream inflows and withdrawals, a sub-basin might experience: i) *no dependency* if scarcity is not affected by upstream inflows, ii) *continuous dependency* if scarcity is avoided thanks to upstream inflows, or iii) *intervened dependency* if scarcity occurs after upstream withdrawals are taken into account. We also distinguish between whether scarcity occurs every year (*persistent scarcity*), or not in every year (*occasional scarcity*). While the precise cut-off between persistent and occasional scarcity could be disputed, this distinction captures the idea that occasional scarcity might be addressed when needed using adaptive measures, whereas persistent scarcity requires permanent arrangements. For the purpose of developing our analytical framework, occurrence of scarcity is determined using commonly used *water shortage* and *water stress* indicators. We are aware that water scarcity can be socially induced where social systems rather than climatic or hydrological factors are determining, disadvantaging groups within society, often those marginalised (Mehta 2013). However, as a first step to assessing global conditions of SBAs we focus on thresholds of physical scarcity following existing studies (Brown and Matlock 2011, Kummu et al. 2010, Porkka et al. 2012). We note that other indicators could be substituted in future work. A summary of these key terms is given in Table 1.



*Table 1 Key terminology used in the analysis and their definitions.*

| Term | Definition |
| --- | --- |
| *No dependency* | A region would not experience water scarcity if they only had access to local runoff |
| *Continuous dependency* | A region would experience water scarcity if they did not have access to upstream inflows |
| *Intervened dependency* | A region experiences water scarcity after accounting for upstream water withdrawals, but would not experience water scarcity otherwise |
| *Local runoff* | Runoff occurring internally within a region |
| *Natural discharge* | Total water availability, calculated as local runoff + upstream runoff |
| *Actual discharge* | Total water availability after upstream withdrawal (natural discharge – upstream withdrawal) |
| *No scarcity* | No scarcity occurs in any year |
| *Occasional scarcity* | Scarcity occurs, but not in every year |
| *Persistent scarcity* | Scarcity occurs every year |
| *Water stress* | Demand driven water scarcity, calculated as use to availability ratio |
| *Water shortage* | Population driven water scarcity, calculated as water availability per capita |

These definitions of upstream water dependency and dependency category form the basis of our quantitative analytical
80  framework. The framework is particularly suited to examining upstream-downstream hydrological linkages, and how they
affect whether or not sub-basins need to be able to cope with occasional or persistent scarcity. The framework is used to
conduct a global analysis that quantitatively distinguishes different experiences of scarcity and dependency on upstream water
at a transboundary sub-basin scale, i.e. parts of basins that belong to different countries. Figure 1 summarises the key ideas of
this paper. Specifically, we aim to answer the following research questions:

85  •  What is the current dependency category of each sub-basin?
    •  How do climate, upstream withdrawals, and local demand influence dependency category? How could dependency
       category change in future?
    •  How might dependency category and its potential future evolution affect negotiations with other sub-basins?

Our analysis is based on modelled water availability and use data (Sect. 2.1). Our method section builds up our analytical
90  framework, defining sub-basins and calculating the different types of water availability (Sect. 2.2.1), interpreting upstream
dependency in terms of water scarcity (Sect. 2.2.2), unpacking determinants of dependency categories (Sect. 2.2.3), and
identifying a typology of possible transitions in dependency category as local demand and upstream water withdrawals change
(Sect. 2.2.4). Applying this method to global transboundary basins, our results first describe dependency categories in 2010
and how they affect the problems faced by the sub-basins (Sect. 3.1). We then describe how the sub-basins fit within the
95  typology, and interpret how this affects negotiation with upstream sub-basins to avoid the need to cope with occasional or
persistent scarcity entirely (Sect. 3.2). We conclude with discussion of opportunities for further work building on and
improving this method and dependency typology.





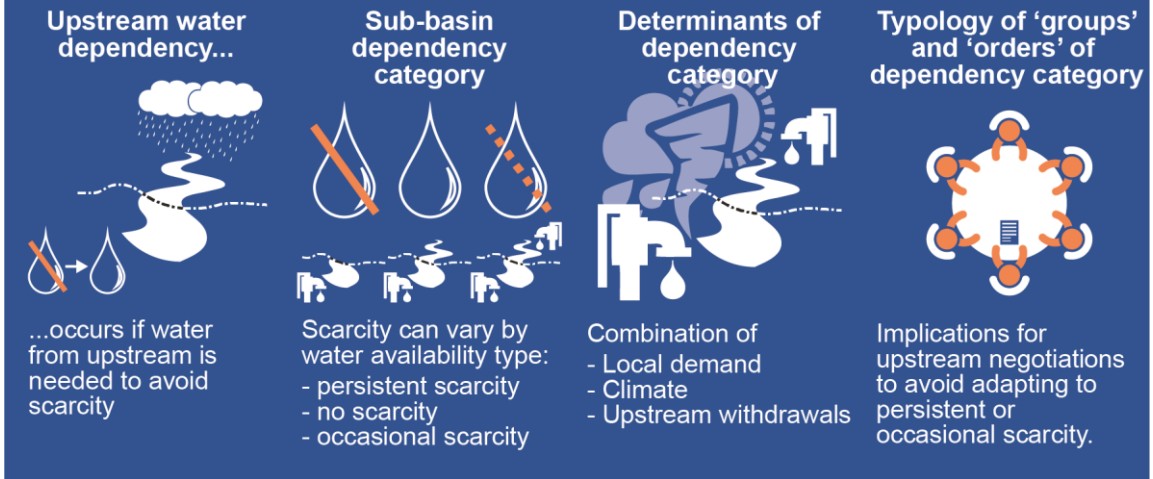

**Fig. 1 Key ideas of this study: our definition of dependency and themes addressed by our research questions.**

## 2. Data and Methods

To operationalise our definition of upstream water dependency, we used the global hydrological model PCRaster Global Water Balance (PCR-GLOBWB) to simulate water use and water availability at grid cell resolution (30 arc-min or roughly 50 km by 50 km at the equator). A basin-country mesh was used to sub divide the transboundary basins into sub-basin areas (SBAs). We then examine differences in the ordering and volumes of available water of the different types in order to provide a first explanation of why dependency occurs. Below we present the data, methods and analytical framework used for the assessment in more detail.

### 2.1 Data

The data used for the study is summarized in Table 2. Runoff and water withdrawals (WWs) were calculated using the PCR-GLOBWB 30 arc-min model (Wada et al. 2011, Wada et al. 2013) . PCR-GLOBWB is a conceptual, process-based water balance model. In brief, it simulates for each grid cell and for each time step (daily) the water balance in two vertically stacked soil layers and an underlying ground water layer, as well as the water exchange between the layers and between the top layer and the atmosphere (rainfall, evaporation and snowmelt) (Wada et al. 2013). Discharge estimates from the model are extensively validated against observations from the Global Runoff Data Centre (GRDC) in existing publications by Wada et al. (2014, 2013).



*Table 2. Datasets used in the study together with their source.*

| Data | Year | Source | Description |
|---|---|---|---|
| *Drainage direction* | - | Döll (2002) | Global grid with 30 arc-min resolution |
| *Runoff* | 1981–2010 | Wada *et al.* (2011, 2013) | Monthly data at global grid with 30 arc-min resolution |
| *Irrigation water withdrawal* | 1981-2010 | Wada *et al.* (2011, 2013) | Monthly data at global grid with 30 arc-min resolution |
| *Industrial water withdrawal* | 1981-2010 | Wada *et al.* (2011, 2013) | Monthly data at global grid with 30 arc-min resolution |
| *Domestic water withdrawal* | 1981-2010 | Wada *et al.* (2011, 2013) | Monthly data at global grid with 30 arc-min resolution |
| *Population density data* | 1981-2010 | Klein Goldewijk *et al.* (2010) | HYDE data set |

Total WW were calculated for each SBA as the sum of three water use sectors: irrigation, domestic and industrial. The water use data for these sectors were obtained from the same model as the discharge simulations (Wada et al. 2011, Wada et al. 2013). Water use estimates have also been previously validated against reported country data, notably FAO AQUASTAT, by Wada *et al* (2011).

To provide an indication of need for water (rather than demand), population density information was obtained from the HYDE
3.2 dataset for each year from 1981 to 2010 (Klein Goldewijk et al. 2010) . The data were first aggregated from 5 arc-min to 30 arc-min resolution and then for each SBA for every year over the 30-yr period.

The 30 arc-min raster dataset DDM30 (Döll 2002) described drainage direction for both surface flow routing in PCR-GLOBWB and definition of upstream-downstream links.

Country boundaries were first rasterized from Natural Earth Admin 0 boundaries (Natural earth 2017). Border cells were then
manually assigned to countries to provide meaningful hydrological relationships. In general, single cell sub-basins were avoided. Cells where country borders follow a river were treated as separate "shared" zones. What we refer to as a "country" raster therefore includes both countries and shared zones.

### 2.2 Methods
#### 2.2.1 Sub-basin definition and calculation of water availability
To explain the methods and analytical framework used for the global assessment, we use the Dnieper River Basin as an example case study (Fig. 2). Dnieper is a transboundary river that rises near Smolensk, Russia and flows through Russia, Belarus and Ukraine to the Black Sea. It is the fourth longest river in Europe. We chose the Dnieper River Basin as an example case study because: i) it has non-trivial but sufficiently easy hydrological connections for illustrative purposes, ii) it includes upstream, middle stream and downstream SBAs, and iii) the water stress levels and downstream dependencies illustrate well the use of
our analytical framework.

SBAs (i.e. sub-basin areas) were defined by breaking up the drainage direction map where it flows across country (and shared zone) boundaries, effectively yielding a mesh of river basin and country boundaries. Upstream-downstream relationships between these SBAs were defined by the flow direction dataset. The construction of the country raster (see Sect. 2.1) ensured that the SBAs provide a meaningful representation of the hydrological system. A country can have multiple sub-basins in order
to capture different flow paths. In general, the drainage direction raster captures major tributaries even if finer details are missing. In the case of Dnieper Basin, Fig. 2 presents the five identified SBAs (DnSBA$_{RU-A}$, DnSBA$_{RU-B}$, DnSBA$_{UA-A}$,



DnSBA$_{UA-B}$, DnSBA$_{BY}$) and the direction of flow between these SBAs. Russia (DnSBA$_{RU-A}$, DnSBA$_{RU-B}$) and part of Ukraine (DnSBA$_{UA-A}$) were identified as the most upstream, Belarus (DnSBA$_{BY}$) as middle stream, and part of Ukraine (DnSBA$_{UA-B}$) as the most downstream (Fig. 2).

Three types of water availability were calculated in each of these SBAs, corresponding to local water (runoff), total inflows including upstream areas (natural discharge), and total inflows after upstream WWs (actual discharge) (see alsoTable 1). We used runoff and WW estimates together with a simplified routing approach to estimate discharge by summing runoff and subtracting WW. This provides an easy to follow abstraction of the problem that emphasises upstream-downstream relationships while ignoring issues of land use change, timing of flows and conveyance losses.

WW for each SBA was calculated separately (referred to as *WW.local*) by summing up the three water use sectors (industrial, domestic and agriculture) and aggregating to SBA scale. Local runoff water availability for each SBA (*avail.local*) was given by its annual runoff. Total water availability (*avail.total*) for each SBA was calculated by summing together the runoff of the SBA and all its upstream SBAs.

        Discharge after upstream WWs (*avail.afterup*) was calculated from the SBA WWs and total water availability. We identified
the entire upstream area for each SBA based on the upstream-downstream hierarchy; i.e. in cases when an SBA has more than one upstream SBA, the total upstream water use is summed (*WW.upstream*). These water use results were then subtracted from natural discharge for the corresponding year, i.e. *avail.afterup = avail.total – WW.upstream*. In some cases, *avail.afterup* in excess of *avail.local* is considered to be fossil ground water or other available water that is not included in the calculation. In these cases, we set *avail.afterup* to be equal to *avail.local* for that SBA.





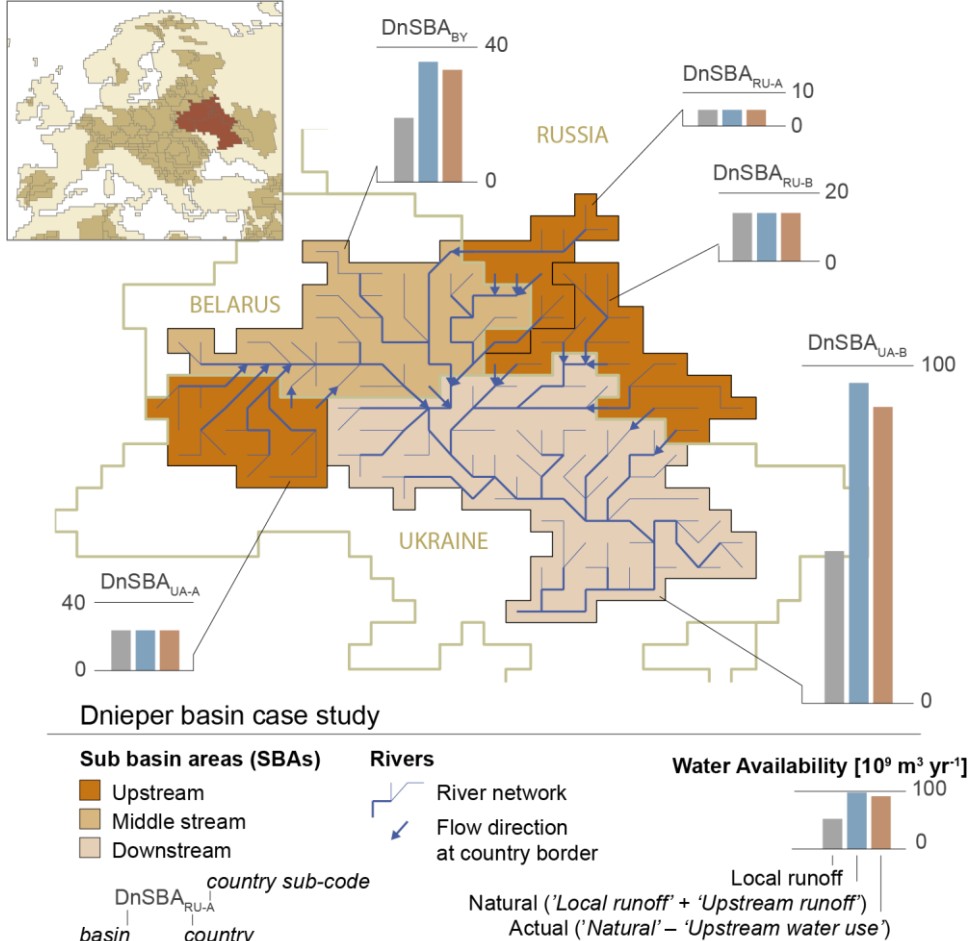

**Fig. 2. Upstream-downstream relationship between SBAs in the Dnieper basin and average simulated water availability for 1981-2010. Drainage network and sub-basin division are based on DDM30 (Döll 2002) and country borders (Natural earth 2017) with additional manual assignment of border cells.**

### 2.2.2    Interpretation of upstream dependency in terms of water scarcity

To understand the concept of upstream dependency, we first looked at the average availability of water (1981-2010) for the SBAs of the Dnieper basin (Fig. 2). Headwater SBAs (DnSBA$_{RU-A}$, DnSBA$_{RU-B}$, DnSBA$_{UA-A}$) obviously have no upstream dependency; the three types of water availability are the same. But in the case of SBAs DnSBA$_{BY}$ and DnSBA$_{UA-B}$, upstream water availability and withdrawals influence water availability. These are the SBAs we are most interested in.

Dependency on upstream water can then be assessed by comparing an SBA's scarcity category across the different water availability types (i.e. local runoff, natural discharge, actual discharge – see definitions in Table 1 ). We calculated scarcity using the water stress and water shortage indices. Water stress refers to impacts from high use of water while water shortage refers to impacts from insufficient water availability per person (Falkenmark et al. 2007, Kummu et al. 2016). The stress indicator was calculated as *WW.local/avail*, the shortage indicator as *avail/population.local*. To determine whether water stress or shortage occurs, we respectively used the thresholds 0.2 and 1000 m³/capita/yr, as defined by Falkenmark et al (2007). Crossing these thresholds respectively means that more than 20% of available water is withdrawn, leading to impacts from high use of water, and that water availability is below 1,000 m³/capita/yr, leading to impacts from insufficient water availability per person, potentially limiting economic development, and human health and well-being (Falkenmark et al. 2007). As





mentioned above, the use of these thresholds is in line with existing studies and while there are notable limitations including that of simplification, we nonetheless utilize them as a first step in understanding upstream dependency.


Annual stress and shortage were calculated using WWs and population for 2010 as:

Equations for water stress with 1) runoff, 2) natural discharge and 3) actual discharge:

1. $\dfrac{WW.local}{avail.local}$ ; 2. $\dfrac{WW.local}{avail.total}$; 3. $\dfrac{WW.local}{avail.total - WW.upstream}$.

Equations for water shortage:


1. $\dfrac{avail.local}{population.local}$ ; 2. $\dfrac{avail.total}{population.local}$; 3. $\dfrac{avail.total - WW.upstream}{population.local}$.

When using average water availability over many years (in our case 30 years), the water scarcity was categorized as
- No scarcity (N) and
- Scarcity (S)

Fig. 3a represents the changes in scarcity for the Dnieper basin under average conditions, shown within the Falkenmark matrix (Falkenmark et al. 2007, Kummu et al. 2016) which shows stress and shortage together. Archetypes in the Falkenmark matrix describe the water scarcity status (corresponding to position on the plot) and where both shortage and stress occur, according to which occurs first (Kummu et al. 2016) .

Though none of the SBAs have any shortage as the per capita water availability (i.e. shortage) has never dropped below 1000
$m^3$ cap$^{-1}$ yr$^{-1}$, DnSBA$_{UA-B}$ would be stressed (S) (as it exceeds 20% threshold value) if it were restricted to its own runoff (Fig. 3a). After accounting for inflows from upstream (natural discharge), the stress level decreased from 0.37 to 0.17 (N) (Fig. 3a). This change in stress category means that DnSBA$_{UA-B}$ is dependent on upstream water to avoid stress. We further see that upstream WW increases the stress level relative to natural conditions (to 0.19) (Fig. 3a & c), but the threshold for stress was not crossed. The stress level without changing the stress category, such that the category of the dependency was not affected;
we have a 'continuous' rather than 'intervened' dependency. In the case of DnSBA$_{BY}$, locally available runoff was sufficient to meet needs, such that upstream water availability and water use does not influence the scarcity category of this SBA (Fig. 3a & c), and it is categorized as 'No dependency'. The dependency category of an SBA can then be summarised using three letter codes representing the scarcity category using runoff, natural discharge and actual discharge respectively: DnSBA$_{UA-B}$ is SNN, DnSBA$_{BY}$ is NNN (Fig. 3a).

So far we have looked at the dependency of the SBAs for average water availability conditions but water availability varies from year to year. To capture this variability, we calculated the water scarcity status using water availability for each year from 1981 to 2010. Scarcity is therefore further categorized as:
- No scarcity (N): no scarcity occurs in any of the years
- Occasional scarcity (O): scarcity occurs, but not every year
- Persistent scarcity (S): scarcity occurs every year

Persistent and occasional scarcity can equivalently be distinguished by identifying whether water scarcity occurs for wet and dry years respectively. While persistent scarcity is obvious because of low water availability in relation to water demand, people may not necessarily be prepared for occasional scarcity, or may need adaptive measures to be actively implemented.

When considering only average conditions, DnSBA$_{UA-B}$ had a continuous dependency, and DnSBA$_{BY}$ had no dependency.
Accounting for occasional scarcity, DnSBA$_{UA-B}$ is described as 'SOO' (Fig. 3b). The dependency is still continuous, but we see that even with upstream inflows, occasional scarcity does occur. DnSBA$_{BY}$ falls under the category 'ONN' (Fig. 3b & d). It would have occasional scarcity if only local water were available while it would be under 'no scarcity' category under natural





and actual discharge (Fig. 3b). This SBA is thus dependent on upstream inflows to avoid occasional scarcity and the dependency is also continuous as the upstream WW does not change the scarcity category. In the cases where scarcity category

under different water availability types remains the same (for example: $DnSBA_{RU-A}$, $DnSBA_{RU-B}$, $DnSBA_{UA-A}$), the SBA is considered to have 'no dependency' (Fig. 3b & d). When this occurs outside headwater SBAs, upstream WW may still affect the frequency and intensity of scarcity, and water availability at which thresholds occurs. Based on the above discussion, the interpretation of possible scarcity in terms of dependency is shown below in Fig. 4.



*Fig. 3 Scarcity and dependency category for the Dnieper basin under average (a and c) and annual variation (b and d) conditions. The Falkenmark matrix (a and b), shows changes in stress and shortage under different water availability (see definitions in Table 1). Inset maps represent the Dnieper SBAs' corresponding dependency categories. Water availability plots for DnSBA$_{UA-B}$ (c and d), show how different water availability types relate to requirements to avoid stress. Scarcity and dependency categories for each SBA for the year 2010 were calculated using a water stress threshold value of 0.2 and water shortage threshold value of 1000 m³/year.*





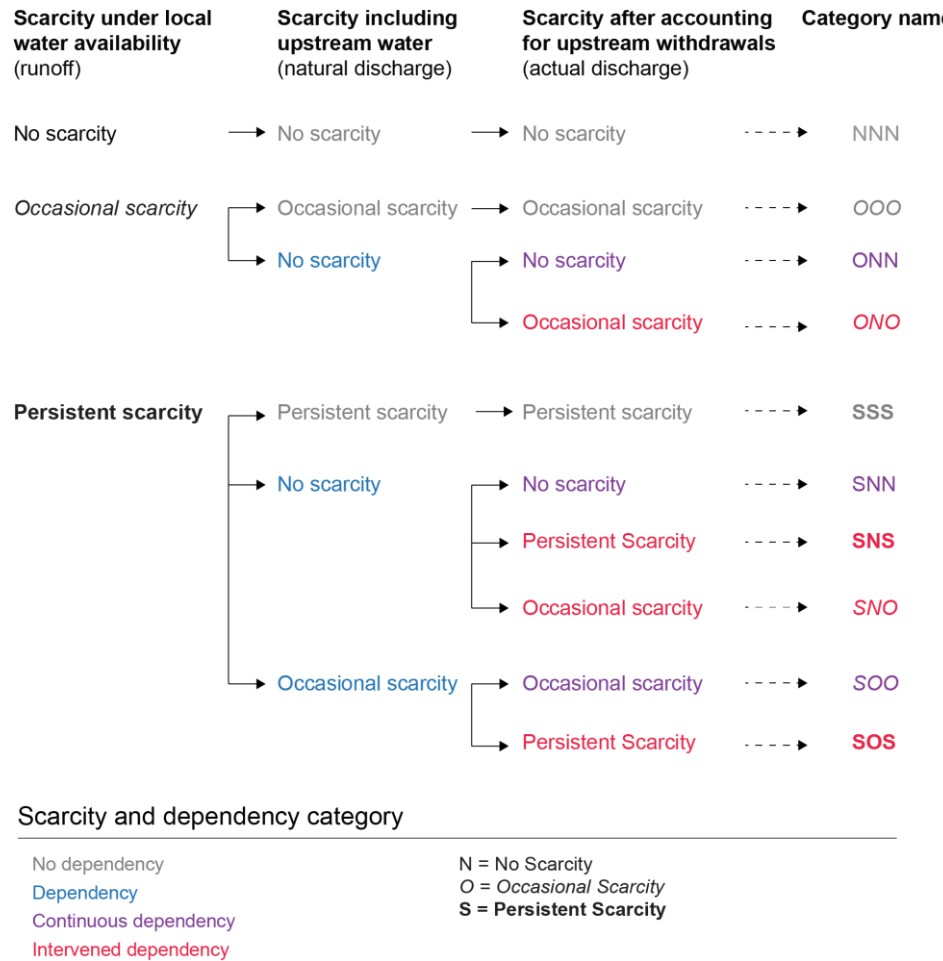

**Fig. 4. Definition of potential upstream water dependency categories. Dependency categories are obtained by summarizing three letter codes representing the scarcity category using runoff, natural discharge and actual discharge respectively (see definitions in Table 1).**

### 2.2.3 Determinants of dependency category

In order to evaluate possible responses to dependency, we need to understand what determines a dependency category and what can be done to achieve or to avoid change. Annual water availability can be thought of as a constraint on the environment in which a society operates. Society is able to influence that constraint, for example by building reservoirs – captured to some extent by the model. However, for a given hydro-climate and state of development, it is useful to think of water availability conditions as a constant. As population and WW increase in a region, the occurrence of shortage, stress and upstream dependency is determined by the volumes of the three types of water availability evaluated in dry and wet years. Specifically, the analysis we have described so far can equivalently be described by saying that as a region's water needs and water use increases, they will cross six thresholds that result in changes in how the system operates. A region will face scarcity or dependency as a result of:

- Insufficient reliable local water availability, available even in a dry year (*avail.min.local*)
- Insufficient less reliable local water, available in a wet year (*avail.max.local*)
- Insufficient reliable dry year discharge, from runoff and possible upstream inflows (*avail.min.total*)





- Insufficient less reliable wet year discharge, from runoff and possible upstream inflows (*avail.max.total*)
- Insufficient reliable dry year discharge after upstream WW (*avail.min.afterup*)
260 - Insufficient less reliable wet year discharge after upstream WW (*avail.max.afterup*)

An SBA would be under 'no scarcity' category when its minimum water availability is sufficient to meet the water use demand in a given year (WW in case of stress and population in case of shortage). 'Occasional scarcity' occurs when an SBA's minimum water availability is insufficient but maximum water availability is sufficient. Finally, an SBA falls into the 265 'persistent scarcity' category when its maximum water availability is insufficient in relation to its water demand.

The advantage of thinking in terms of thresholds is that we can reason about how scarcity and dependency might change in future. Changes in upstream water use might then lead to crossing the scarcity threshold. In this paper, we focus on the effect of increasing or decreasing local demand and upstream water use, leaving changes in water availability to future work.

Fig. 5 shows the ordering of thresholds for the Dnieper's DnSBA$_{UA-B}$, and how the shortage and stress categories varies as 270 demand changes, considering the case without changing upstream WW for the time being. To allow comparison, water availability, population, and withdrawal are all expressed as percentages respectively of *avail.max.total*, carrying capacity (*avail.max.total*/1000) and sustainable yield (*avail.max.total* * 0.2). Currently the DnSBA$_{UA-B}$ is in the 'SOO' category for stress and 'NNN' category for shortage (Fig. 5). It has a continuous dependency (for avoiding stress), as it is experiencing occasional rather than persistent scarcity thanks to upstream inflows. If the water use in DnSBA$_{UA-B}$ were to increase to a level 275 where the maximum wet year inflows after upstream WW (*avail.max.afterup*) would not be enough to meet the water use demand, the SBA would next transition from SOO to SOS category. Thus, with the increase in demand, the dependency category (e.g. 'SOS') would change based on the thresholds it crosses and ultimately the basin would become SSS, indicating that an SBA would be under scarcity every year under each available water type considered. The same thing would happen with shortage as the population increases (Fig. 5). Over time, this change in dependency category could go forward and 280 backward as water demand of the SBA increases or decreases.

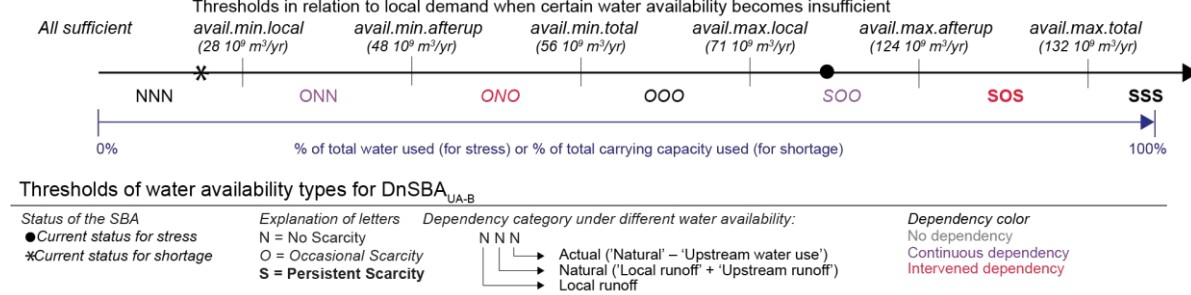

*Fig. 5. Thresholds of water availability types for Dnieper's sub-basin area DnSBA$_{UA-B}$ and its dependency category (for both stress and shortage) in 2010. See definitions of used terminology in Table 1*

### 2.2.4 Typology of possible transitions in dependency category

So far, we have conceptualised change in dependency category in the context of a fixed set of water availability thresholds, obtained directly from estimated water availability volumes. The order of thresholds determines the transition in dependency category as local demand increases or decreases. These thresholds are, however, expected to change depending on climate and upstream WW. It is therefore useful to build a typology of possible orders of thresholds to better understand potential future implications and changes to use.

We begin from the observation that the scarcity thresholds are (partially) naturally ordered. In terms of how they are defined, reliable and local water necessarily becomes insufficient before less reliable and upstream water availability types respectively:





*local ≤ afterup ≤ total*, and *min ≤ max*. Given that these constraints only partially determine the order in which scarcity and upstream dependency is experienced, SBAs can be categorised in four different groups and six distinct orders as follows (Fig. 6):

*Headwaters* (Group 1) are the simplest case. Given they are the most upstream SBAs, they rely solely on local runoff, and have a single ordering of thresholds (*Order I*). Increases in an SBA's demand cause transition from 'no scarcity' to 'occasional scarcity' and finally to 'persistent scarcity' category. Decrease in demand would have the opposite effect (Fig. 6).

The next distinction between orders separates out SBAs with *no reliable upstream support* (Group 2 in Fig. 6). In this group, the volume of reliable dry year inflows from upstream areas (avail.min.total) determines ordering of the thresholds. For Group

2 SBAs, discharge in a dry year is lower than wet year local runoff, i.e. *avail.min.total ≤ avail.max.local*. In natural conditions, upstream inflows enable other SBAs to increase their demand beyond their wet year local runoff without experiencing any scarcity (SNN), though scarcity may still occur with high upstream WW (ONO). Group 2 SBAs do not even have this possibility, and instead experience occasional scarcity without upstream intervention (OOO). Group 2 only has one possible order of thresholds (*Order II*). We see in Fig. 6 that transitions in category can occur due to increases in either local demand

or upstream WW where a dependency is present (ONN -> ONO, SOO -> SOS), but only due to local demand where there is no dependency or an intervened dependency (e.g. NNN -> ONN, ONO -> OOO). Conversely, transitions occur due to decreases in either local demand or upstream WW from an intervened dependency category (ONN <-ONO, SOO <-SOS), but only due to local demand in other cases. This is a result of our definition of dependency and holds true for all transitions.

In the remaining two groups, the ordering of thresholds is linked to whether upstream WW exceeds 1) the reliable upstream

support (*avail.min.total - avail.max.local*), and 2) natural discharge variability, specifically, the range of downstream natural discharge (*avail.max.total - avail.min.total*) (Group 3 and 4 in Fig. 6). The level of upstream WW has implications for whether inflows help to avoid occasional and persistent scarcity. If upstream WW exceeds natural reliable upstream support, such that *avail.min.afterup ≤ avail.max.local*, inflows no longer forestall *occasional* scarcity. The SBA experiences the transition ONN -> ONO rather than ONN -> SNN (Group 2 and 3 in Fig. 6). If upstream WW exceeds discharge variability, then inflows after

upstream WW even in a wet year will be lower than natural inflows in a dry year (*avail.max.afterup ≤ avail.min.total*), meaning that inflows no longer forestall *persistent* scarcity. The SBA faces the transition SNO -> SNS rather than SNO -> SOO (Group 2 and 3 in Fig. 6).

The two groups can therefore be differentiated by whether reliable support is greater or less than natural discharge variability. We define Groups 3 and 4 as having low and high reliable support respectively (see Fig. 6). Group 3 has reliable support <

discharge variability. Group 4 has reliable support > discharge variability. The combination of the two upstream WW conditions yield four orders: III, IVa, IVb, V. Orders III and V are cases where upstream WWs are respectively higher and lower than both reliable support and natural discharge variability. Order IVa occurs when upstream WW are higher than reliable support and lower than natural discharge variability, which is not possible in Group 4, and therefore only occurs in Group 3. Conversely, order IVb occurs when upstream WW is lower than reliable support and higher than natural discharge

variability, and only occurs in Group 4.



*Fig. 6. Typology of groups and orders of possible transitions in dependency category, as local water demand or upstream water withdrawals (WW) increase/decrease. Upstream WWs decrease the downstream water availability, while local water demand increases the pressure on available resources and thus, their impact on water scarcity differs given the group, order and status of a given sub-basin area (SBA). See definitions of used terminology in Table 1.*

## 3.    Results and interpretation of global analysis

The analysis was applied to 246 international transboundary basins to understand the dependency category of these basins and possible future transitions, using water use and population data from 2010.

### 3.1    Identified sub-basins under different dependency categories

The 246 transboundary basins were divided into 886 SBAs based on country borders (as well as shared zones along those borders). As shown in Table 3 , in the case of stress, most SBAs had no dependency in 2010 (88%, 782 SBAs), though a substantial number (88 SBAs) did have a continuous dependency – upstream WW does not change the scarcity categories. In





these cases, upstream users are not responsible for the occurrence of downstream scarcity, but might intensify scarcity (see Discussion). In total 16 (2%) SBAs out of 886 SBAs are identified where dependency was 'intervened', meaning that upstream water use has changed stress category (Table 3). The picture was similar for shortage, with 72 SBAs with continuous dependency and only 7 with intervened dependency. Upstream water withdrawals only rarely play a role in causing low water availability per capita.

*Table 3 Number of sub-basins under different dependency categories in the year 2010.*

| Dependency category | | Stress | | | Shortage | | |
|---|---|---|---|---|---|---|---|
| | | No of sub-basins | | Population ($\times 10^6$) | No of sub-basins | | Population ($\times 10^6$) |
| **No upstream dependency** | NNN | 595 | | 1102 | 711 | | 2055 |
| | OOO | 109 | 782 (88%) | 324 | 63 | 807 (91%) | 105 |
| | SSS | 78 | | 403 | 33 | | 133 |
| **Continuous Dependency** | ONN | 43 | | 75 | 59 | | 242 |
| | SNN | 22 | 88 (10%) | 249 | 4 | 72 (8%) | 65 |
| | SOO | 23 | | 608 | 9 | | 157 |
| **Intervened Dependency** | ONO | 2 | | 5 | 6 | | 35 |
| | SNS | 0 | | 0 | 0 | | 0 |
| | SNO | 9 | 16 (2%) | 8 | 1 | 7 (0.8%) | 0 |
| | SOS | 5 | | 18 | 0 | | 0 |
| Total | | | 886 | 2792 | | 886 | 2792 |

'No dependency' is observed in 88% of cases for stress and 91% of cases for shortage (Table 3). It is worth noting that scarcity can still be experienced without a dependency – it simply means that current upstream inflows (and WWs) do not influence whether scarcity occurs. There is not currently a problem with relationships with upstream basins, but to plan ahead, we need to understand how the situation could evolve, as will be discussed in Sect. 3.2.

'Continuous dependency' is observed for both stress and shortage mostly in Africa, Europe, some parts of Southeast Asia, and North America (Fig. 7a, b). Continuous dependency means that relationships with upstream basins are important. Many SBAs, in which currently no scarcity is observed (Fig. 7), are actually suffering from upstream dependency. If inflows decrease sufficiently due to upstream WWs, scarcity could occur, or become persistent rather than occasional. In these SBAs, this does not yet happen, though upstream WWs may be influencing the frequency and intensity of scarcity, and the level of development (population or use) at which thresholds occurs. Therefore, understanding of how the situation can evolve is needed to know how to manage the relationship with upstream water users.

'Intervened dependency' occurred mostly in different parts of central Asia for both stress and shortage (Fig. 7a, b). Intervened dependency indicates that there is a potential for tension with upstream water users over water allocation. There would be no scarcity or only occasional scarcity if it were not for upstream WWs – but reducing local water needs or WWs could similarly avoid shortage or stress. As a result, in these SBAs water management has become an uncoordinated competition between the upstream and the downstream region. Such situation is already evident in case of Central Asia (Dukhovny 2014). However, understanding of the evolution of the situation may show that small decreases in local or upstream WWs may not be sufficient to avoid scarcity or dependency. It may be necessary to find means to reduce needs or adapt to impacts from high water use.




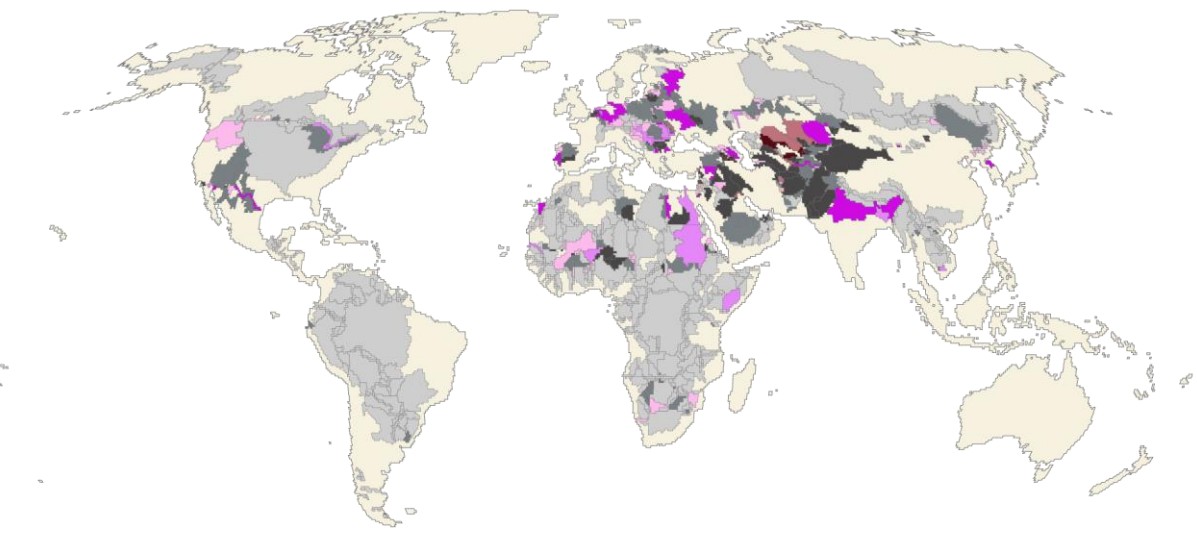

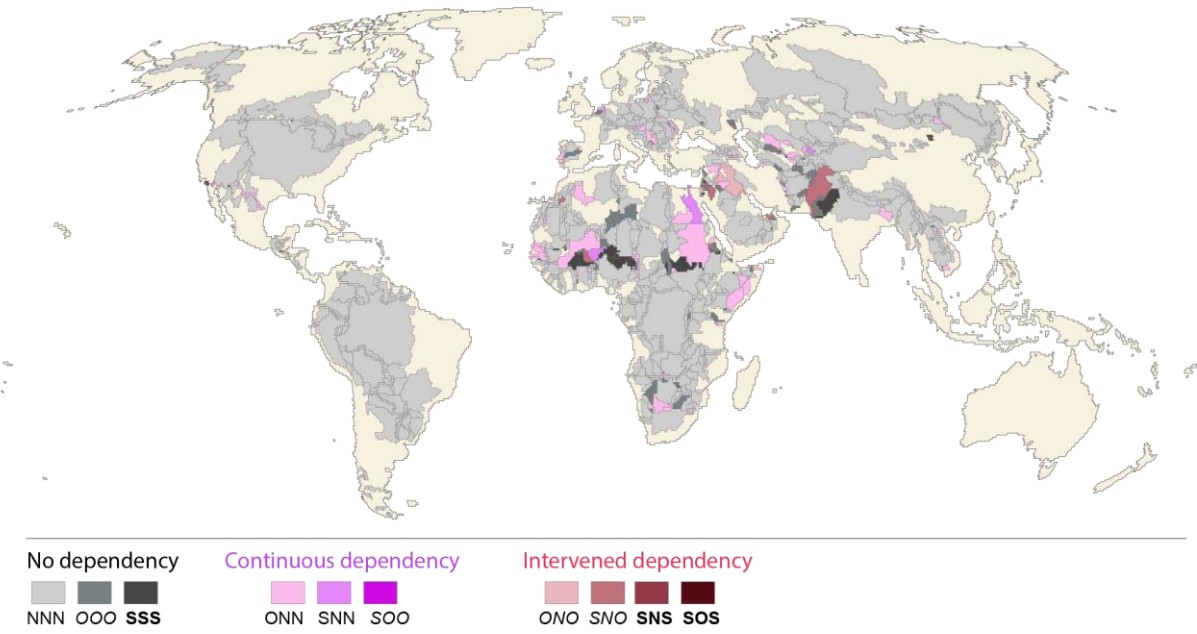

**Fig. 7. Dependency categories for each SBA for the year 2010 using a) Water stress threshold value of 0.2 b). Water shortage threshold value of 1000 cubic meter/year.**

### 3.2    Possible future transitions and implications

Rather than trying to predict the future, we look at the potential future transitions that can occur in each sub-basin. Fig. 8 shows the scarcity threshold orders and the dependency category of SBAs under these orders for both stress and shortage, for current climate and corresponding upstream WWs (1981-2010). Fig. 9a in turn maps where the different orders occur.





Understanding potential transitions as well as current dependency category allows us to make suggestions for how negotiation in upstream-downstream relationships might be influenced. Headwaters obviously have no upstream dependency (Group 1, *Order I*) – the management of their relationships will be guided by how they influence downstream SBAs.

Dependency categories in all orders except *Order I* begin with NNN-ONN and end with SSS (Fig. 8). The need to engage with
upstream water users begins with an increase in local water demand, creating a situation where scarcity depends on upstream withdrawals. From that point on, an SBA's experience of water scarcity depends on the group and order to which they belong, which is determined by the climate-dependent level of reliable upstream support as well as the level of upstream withdrawals. However, if local demand increases sufficiently, any SBA can find itself in a situation of persistent scarcity that is not dependent on upstream (SSS). If an SBA wishes to avoid the need to cope with occasional or persistent scarcity, negotiation
with upstream is only relevant when local demand is sufficiently low (but negotiation may still affect the severity of scarcity and its impacts).

Revisiting the results of Table 3 in this context, the need for negotiation in transboundary basins could become a much greater issue in future. In terms of stress, 72% of SBAs (638) are still in the preliminary stages (NNN & ONN), and only 8.8% (78) have reached the final stage where persistent scarcity is accepted as a fact of life. Shortage is even less developed, with 87%
of SBAs (770) at NNN or ONN, and 3.7% (33) at SSS. There are more cases of SSS in Group 2 than 3 and 4, likely as a result of lower reliable upstream support. The intricacies of groups and orders are likely to become relevant to these SBAs in future, if local water demand increases.

A first key pattern is whether an SBA transitions from ONN to ONO or to SNN (Fig. 8). In the case of ONO, the SBA experiences occasional scarcity for the first time, and this would not (yet) have occurred without upstream withdrawals (it is
an intervened dependency). In the case of SNN, the SBA escalates their dependency – persistent scarcity would occur without upstream inflows. In Group 2, ONO is unavoidable (Fig. 8). The lack of reliable upstream support means that negotiation with upstream can only tweak how much local demand can increase before occasional scarcity will occur. However, in Groups 3 and 4, SNN could be achieved if upstream withdrawals are kept sufficiently low. Currently, for stress most SBAs fall in SNN, and for shortage the SBAs are approximately equally distributed. Further work would be needed to understand why the
upstream-downstream water allocation turned out that way. In Group 2, very few SBAs are in ONO, likely because minimum flows are relatively low (less than *avail.max.local*).

A second key pattern relates to transitions between no, intervened and continuous dependencies. Continuing increases in local demand in Order II (Group 2) mean that the SBA reaches states where there is no dependency on upstream water, i.e. OOO and SSS. The preceding continuous (ONN, SOO) and intervened dependencies (ONO, SOS) are only intermediate states.
Keeping upstream WWs low delays the occurrence of occasional scarcity, but it would occur regardless of upstream WWs if local demand continues to increase. If the intention is for local demand to continue increasing, negotiating to avoid occasional or persistent scarcity is of limited use. Negotiation could instead be focused on timing, frequency and intensity of scarcity.

A similar effect occurs in Order IVa and Order V. Transitions from SNO to SOO mean that occasional scarcity caused by upstream withdrawals would occur even with natural discharge as local demand increases. Again, if a region intends to develop
its resources far enough, it will eventually need to be able to cope with occasional scarcity anyway. These no-dependency transitions provide anchor points in negotiation. An upstream SBA can argue that its downstream neighbour would eventually need to adapt to occasional scarcity anyway. There are a large number of SBAs in these orders (Fig. 8; Fig. 9a). There is also a large potential for latent conflicts due to the low level of upstream development (and hence potential for growth) and high level of downstream development before scarcity emerges.

In Order III and Order IVb, all transitions from ONN or SNN onwards are to intervened dependencies. Downstream SBAs can argue that they would not need to deal with occasional/persistent scarcity, were it not for upstream withdrawals. In the case of Order IVb, a counter-argument is that expansion of local demand beyond SNN suggests that the downstream SBA is in fact choosing to have to deal with intervened dependencies. If upstream demand is stable, this means that the conflict is effectively



of their own making. There are relatively few cases in Order III and none in Order IVb. The most evident cases are in Central
Asia (Fig. 8; Fig. 9a) where downstream SBAs are highly dependent on upstream inflow and their dependency is intervened
by upstream WW (Fig. 7). Further investigation would be needed to determine whether these orders are being avoided by
coincidence, or by effective negotiations.

Transitions from orders V->IVa->III and V->IVb->III unsurprisingly increase the need for negotiation with upstream SBAs.
Common ideas in water allocation are relevant. The idea of precedence is raised when we look at transitions as demand
increases, keeping upstream WWs fixed. Scarcity, continuous dependencies and intervened dependencies only emerge as
problems when local demand crosses a threshold. This gives the impression that it is the local user that is responsible for the
new problem, even though it may simply be that they are late to the game. Considering both local and upstream demand
simultaneously, instead, emphasises a negotiation on more equal terms, where both local and upstream users are responsible
for the scarcity outcomes occurring in the downstream region. The precedence paradigm is visible in prior appropriations
regimes in USA, while negotiated allocations are arguably implemented by water markets in Australia and elsewhere (Grafton
et al. 2011). Even in negotiations, existing water needs and WWs are often taken into account, including at an international
level.

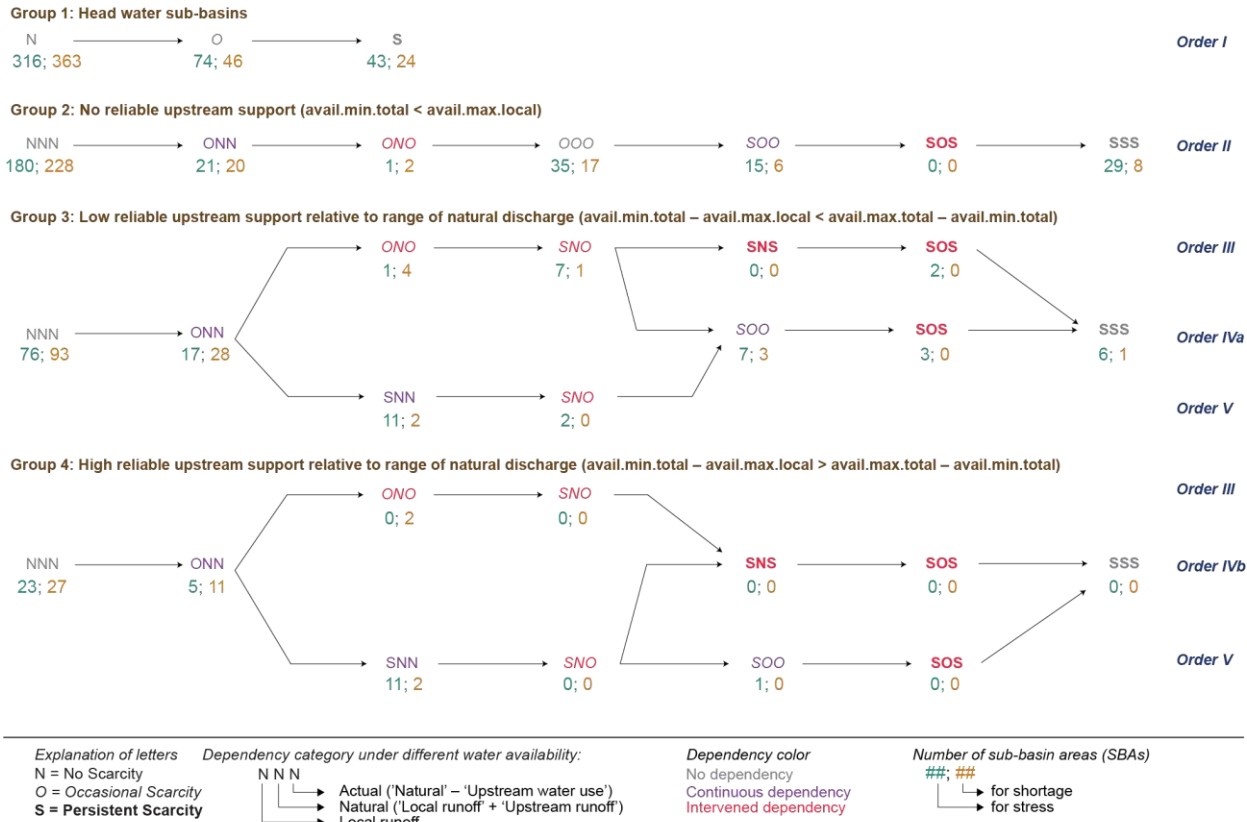

**Fig. 8 . Current (2010) dependency category of SBAs in the context of their order and group.**






Fig. 9. a) Sub-basins under different dependency ordering b) Sub-basins under different groups. The first defines possible future transitions if local demand changes with upstream withdrawals staying at current levels. No sub-basin was found under order IVb. The second defines what transitions are possible with changes in both local demand and upstream withdrawals, with current climate.




## 4. Discussion

In this analysis, transboundary water dependency was examined through the idea that a sub-basin is dependent on upstream inflows if it requires those inflows to avoid water scarcity (e.g. stress, shortage) and associated impacts. We aimed to address three research questions. Firstly, we identified the current dependency category of each sub-basin. Examining occurrence of scarcity with different types of water availability allows classification of ways in which upstream and downstream SBAs are dependent on each other (Sect. 2.2.2, 3.1). To answer the second question, we further developed the analytical framework by explaining how climate, upstream withdrawals and local demand influence the dependency categories (Sect. 2.2.3). This in turn allowed us to describe a typology of possible transitions in dependency categories (Sect.2.2.4). The typology is built on the observation that different types of water availability define tipping point thresholds involving change in scarcity categories, and that these thresholds can be ordered differently depending on climatic characteristics and upstream withdrawals. The third question involved exploring how dependency category and its evolution might affect negotiations with other sub-basins. We have provided a first step in this direction by proposing implications arising from key patterns in the typology (Sect. 3.2). Our work, however, highlights that negotiation to avoid needing to cope with occasional or persistent scarcity is only part of the issue. Negotiation among the riparian countries will eventually turn to discussion of intensity and frequency of scarcity, and the level of demand at which it occurs. Other existing work also distinguishes different types of rivers and basins to help understand why some riparians on international rivers have been able to successfully negotiate treaties and others have not, taking into account, for example, civilization, size of population, GDP, upstream-downstream relationship, and asymmetries in economic and political power among riparian states (Delbourg and Strobl 2012, Song and Whittington 2004, Wolf et al. 2003). Increasing water scarcity has been identified as a risk factor, but has not previously been systematically explored in terms of upstream dependency. Our dependency category typology complements this existing work, and relations to other typologies could be explored in future.

One of the main advantages of our analytical framework, compared to existing knowledge, is that it highlights the possible 'hidden' dependency of upstream water, which has not been assessed in these terms before. Previous studies on transboundary river basins identified clear evidence of the impacts of upstream water use to downstream water availability and water scarcity level (Al-Faraj and Scholz 2015, Munia et al. 2016, Nepal et al. 2014, Veldkamp et al. 2017). It has already been found that about 0.95–1.44 billion transboundary people are under stress because of local water use, while upstream water use increased the stress level by at least 1 percentage-point for 30–65 sub-basins, affecting 0.29–1.13 billion people (Munia et al. 2016). Our analysis provides a different view of the issue by revealing that 932 million people (33% of the total transboundary population) are dependent on upstream water to avoid possible stress because of their own water use and 464 million people (17% of the total transboundary population) are dependent on upstream water to avoid possible shortage (Table 2). Along with previous work, including broader discussion of hydro-political dependency (Brochmann and Gleditsch 2012, Giordano and Wolf 2003, Gleick 2014, Jägerskog and Zeitoun 2009, Mirumachi 2015, Mirumachi 2013, Wolf 1998, Wolf 1999, Wolf 2007) , our analysis reinforces that it is important to consider how upstream inflows help to avoid water scarcity and affect water use and management.

The analytical framework itself is admittedly relatively complex, and much of our theoretical work has involved trying to unpack it and make it accessible to a sufficiently broad audience. The categories of dependency and order discussed in the paper emerge logically when looking at whether water scarcity occurs occasionally or persistently with local runoff or with upstream inflows. The ultimate aim is relatively simple. The typology developed in the analysis aims to describe how the upstream dependency evolves, while emphasizing relationships with upstream WW. Ordering helps to explain the current situation of the SBA in question and the reason behind the situation. Ordering also provides a framework to anticipate an SBA's possible future experience of scarcity and dependency, as local demand and upstream WW increase or decrease. It further helps to manage the risk of water scarcity based on preparedness rather than a crisis approach.

The main emphasis of the paper was the development of the analytical framework to understand the dependency categories and their orders. In this study, we provide the first attempt to link the dependency order to management strategies that could be taken to ease the possible scarcity situation. In future studies, it would be important to develop this further to identify more





concrete strategy options, for example integrating with existing work on infrastructure development (e.g. storage using reservoirs or aquifers) (Daneshmand et al. 2014) and treaty formation (Brochmann et al. 2012, Dinar et al. 2011, Kliot et al. 2001). In connecting to management, the relevance of frequency of scarcity could be further examined in order to provide a more meaningful distinction between occasional and persistent scarcity: at what frequency of scarcity do management options

need to be implemented permanently rather than only adaptively? e.g. trading of temporary vs permanent water allocations (Bjornlund 2003) . The analytical framework could also be applied to water availability and demand scenarios based on future climate change (Representative Concentration Pathways, RCPs) (Van Vuuren et al. 2011) and Shared Social Pathway scenarios (SSP) (O'Neill et al. 2014) to identify what outcomes may be plausible within the full typology described in this paper. In doing so, the scarcity criteria could also be revisited. The analysis can be integrated with the concept of 'adopting tipping

points (ATP)' to understand what strategies are needed (Kwadijk et al. 2010) to cope up with the scarcity status. Additional insights may be gained using other thresholds and/or other water scarcity indicators, such as food self-sufficiency (Gerten et al. 2011, Kummu et al. 2014) or sustainability of water withdrawals (Wada and Bierkens 2014). Further, our estimation of water availability does not take into account potential industrial or domestic pollution in upstream parts of a basin, which might make water unusable for irrigation or domestic purposes (Thebo et al. 2017). Our method was applied here at the basin scale,

considering only international transboundary basins. It can, however, also be applied to understand the dependency at different scales to interpret, for example, more localised water dependencies, e.g. between states within countries (Garrick 2015) .

## 5. Conclusions

In this paper, we aimed to explore the relationships between SBAs (i.e. sub-basin areas) of global transboundary river basins, in terms of dependency of downstream on upstream inflows to meet their needs and avoid water shortage and stress.

Transboundary water dependency was examined through changes in scarcity category across different types of water availability (runoff, naturalized discharge and actual discharge). The evolution of scarcity and dependency of an SBA for a given climate can be categorized into different orders of dependency categories along which a SBA progresses as its water use or water availability changes. This framework helps to understand the dependencies on which the SBA in question relies in order to avoid water stress and shortage, and what may happen if the demand (or population) increases further. Understanding

of the dependency category of an SBA may have policy implications regarding negotiation and redistribution of water among stakeholders, which may assist in improving water management.



**Acknowledgements**

The work was financially supported by Academy of Finland funded project WASCO (grant no. 305471), Emil Aaltonen
Foundation funded project 'eat-less-water', *Maa- ja vesitekniikan tuki ry*, and Academy of Finland SRC project 'Winland'.
We highly appreciate the help and support of Prof Olli Varis and our other team members.



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
