# Peer review of "How downstream sub-basins depend on upstream inflows to avoid scarcity: typology and global analysis of transboundary rivers"

_Hydrology and Earth System Sciences, 2017_

## Referee Comment (RC1) · E. Mostert (Referee) · 28 Sep 2017

The paper analyses the different types of dependency of downstream sub-basin areas (SBAs) on upstream SBAs, and provides a global overview of the types of dependency in 2792 SBAs. That is potentially interesting. However, the concepts used are not completely convincing and they are not always used consistently. Moreover, parts of the paper are overly complex.

1. The paper distinguishes three types of dependency of downstream SBAs on up-

stream SBAs: no dependency, continuous dependency, and intervened dependency. According to the definition in table 1, no dependency means that for the SBA local runoff is sufficient. Yet, in table 3 and elsewhere other cases are qualified as "no dependency" as well: SBAs that experience occasional or persistent scarcity even if they were to receive all natural runoff from upstream. That is not consistent. Moreover, in the latter cases dependency on upstream SBAs is actually high: there is already little water, and every extra drop that is used upstream results in even less water for the downstream SBA. For these cases I would introduce a fourth type of dependency, which might be called "absolute dependency".

2. Continuous dependency is defined in two slightly different ways: on p. 2 as scarcity that is avoided thanks to upstream inflow, and in table 1 as a region that would experience scarcity if it did not have access to upstream inflows. The latter formulation seems to include actual water scarcity as a result of upstream water withdrawals ("intervened dependency"). This is probably an inaccuracy. More problematic is that "continuous dependency" covers very different cases: cases where upstream inflow is so big that downstream scarcity is just a theoretical possibility, and cases where downstream scarcity is a serious threat because of concrete plans to increase upstream withdrawals (or plans to increase water use downstream or the effects of climate change). It would be good to distinguish between these situations, at least in the discussion. In addition, I would replace the term "continuous dependency" by for instance "potential dependency" because there is no dependency if scarcity is just a theoretical possibility. "Intervened dependency" could then become "actual dependency."

3. To calculate water availability in the different SBAs, the paper uses the PCR-GLOBWB model. It is not clear to me whether and how return flows were taken into account. Especially for industrial and domestic water withdrawals these can be significant.

4. The authors distinguish between occasional scarcity - scarcity that occurs only in a dry year - and persistent scarcity - scarcity that also occurs in a wet year. They do not

define wet year and dry year. What return period is used? And why not use instead of wet year average year? Wet year water availability seems to me a very shaky basis for water scarcity management. Please reflect on this.

5. My most important concern is that the typology of possible transitions in dependency category is very complex and it is not clear to me how useful this typology is. What downstream SBAs need to know is how total water availability may change as a result of climate change, how water use upstream may develop, and what their own plans and expectations are concerning water use in their own SBA. On that basis they can anticipate (an increase in) water scarcity and decide to enter into negotiations with upstream SBAs. They do not need and probably would not benefit from a full overview of groups and orders of possible transitions in dependency category.

6. Finally three suggestions for the presentation. First, the different formulation in line 255 can be simplified and made more uniform by removing "reliable" and "less reliable" and putting "dry year" and "wet year" (or "average year") always at the same place. Secondly, if no scarcity is N, occasional scarcity is O, why not use P for persistent scarcity? And thirdly, in table 4 the order in every column could be the same, e.g. always first no scarcity, then occasional scarcity, and then persistent scarcity.

In conclusion, I can see a publishable paper on different types of dependency that analyses the global situation, but it still needs a lot of work. I am not convinced of the relevance of the complex typology of possible transitions.

---

## Referee Comment (RC2) · E. Ansink (Referee) · 15 Oct 2017

**Review of hess-2017-537**
**October 12, 2017**

This paper is on the interconnectedness of water withdrawals and water scarcity in transboundary basins. A method is presented to formally analyze this interconnectedness. This method is subsequently applied to a global hydrological model. The results show that (in my interpretation), interconnectedness is generally low. The implication is that water scarcity is mostly a local problem, which is new to me. My overall assessment is that this paper is a solid piece of work with a new result that has changed my perspective on the management of transboundary rivers, and I thank the authors for this contribution.

I do have some comments. Most of them relate to a lack of precision in the use and application of definitions and terminology. My comments may have implications for (the presentation of) both analysis and results.

**Major comments:**

1. The definitions of dependency in Table 1 are not mutually exclusive, although they should be. A visual representation of the authors' definitions and my proposal to adjust them are displayed in Figure 1 below. Adjustment would probably have some consequences for the analysis, which I hope/expect are easy to incorporate. If not, one simplification would be to merge the 'dep' and 'oops' categories. Perhaps the resulting categorization is the one intended by the authors. An even simpler, and perhaps more relevant categorization is to not only merge 'dep' and 'oops', but also merge 'no dep' and 'still no dep'. Results and insights will stay the same but the presentation will be easier.

[Figure]

Figure 1: Dependency. Top plot according to Table 1. Bottom plot proposed by reviewer.

2. The terms used in Table 1 are not used consistently in the text.

   - The authors use the terms runoff and discharge interchangeably (even in Table 1), which is confusing.

   - The terms 'water withdrawals', 'need' and 'demand' are introduced as different concepts (L129) but they lack proper defitions. Perhaps 'demand' should be replaced by 'quantity demanded', which is something different, or 'use'.

   - In L164 the term 'discharge after upstream WW' is used where authors probably refer to 'actual discharge' from Table 1. In the same paragraph, variable '*avail.afterup*' is

introduced to reflect the same term, so that we now have three terms for the same concept. More variables are then introduced that face the same problem. This is really confusing and obscures the line of argumentation in the main text.

3. Another comment on Table 1. The order of presentation is illogical and should be reversed. Start with water stress/shortage, which you need to understand scarcity, then runoff/discharge, both of which you need to understand dependency. A more bold suggestion is the following. Since you assign variable names to some of the terms in Table 1, it would perhaps be transparent to introduce a formula for dependency (with shorter variable names), which would make it much easier to understand the definitions. For example, if $q_i$ denotes water use in sub-basin $i$, $e_i$ denotes local runoff, and $P_i$ denotes the set of $i$'s predecessors (i.e. sub-basins strictly upstream of $i$) we can write:

   - $\hat{e}_i = e_i + \sum_{j \in P_i}(e_j)$ as the total water available after upstream withdrawal;
   - $\bar{e}_i = e_i + \sum_{j \in P_i}(e_j - q_j)$ as the total water available after upstream withdrawal.

   Subsequently, when we denote $x_i$ as the measure of water needed to avoid water scarcity (be it from stress or shortage), we have:

   - $x_i \leq e_i \quad \rightarrow \quad$ no dependency;
   - $e_i < x_i \leq \hat{e}_i \quad \rightarrow \quad$ still no dependency (see bottom plot of Figure 1);
   - $\hat{e}_i < x_i \leq \bar{e}_i \quad \rightarrow \quad$ dependency.

   These formalizations of the definitions may also assist in discussing e.g. the typology of dependence categories in Section 2.2.4. I realize that I might be pushing this point too far. If this is the case then at least sharpen and streamline the definitions and terms used in the paper in a consistent way.

4. While you mention treaties on transboundary river water in the discussion, they seem to be ignored in the analysis. Dependencies may not be as severe when they are mitigated by treaties that provide security of continuous upstream inflow. Such treaties may even feature well-designed (flexible) sharing rules able to mitigate the impacts of e.g. climate change. We could even have reversed dependency when a treaty stipulates that local runoff should be shared with downstream riparians. In this case, even if local runoff would be sufficient to satisfy demand, the upstream country would be dependent on the downstream country(/-ies). An example would be Ethiopia's position in the Blue Nile basin.

**Minor comments:**

1. L53: Please define 'sub-basin' upon first use.

2. L53: 'experiences' → 'may experience'.

3. L55: 'Parts of basins' do not 'realise' much.

4. Figure 1 duplicates Table 1 and can be removed.

5. L131: The 30yr period is introduced here without any explanation. Why? And how?

6. L155–159: Are return flows accounted for?

7. L165–166: What if an SBA has multiple downstream SBAs? Possibility of double-counting.

8. L198–199: What happened to 'persistent' and 'occasional' from Table 1?

9. Figure 4: the color code categorizes SSS as featuring 'no dependency' which seems incorrect. In general, I would say that any setting where there is scarcity under actual discharge (i.e. after upstream water use) should be coded as 'intervened dependency', since the upstream water use exacerbates the downstream scarcity. I realize that the authors would probably say that this is a case of 'no dependency' because there would also be scarcity withouth upstream water use, but that is a semantic argument since scarcity is coded here as a binary variable.

10. The term 'ordering' and the arrows used in Figures 6 and 8 suggest that sub-basins can only develop in one direction, namely from good (NNN) to bad (SSS). You may want to present a more nuanced story, explaining under what circumstances this tendency may be reversed.

11. Figure 6 is presenting too much at the same time. From the text I understand that there is a natural ordering, but I do not see the added value of presenting all possible pathways through these orders. Same of course for Figure 8. Can you somehow summarize this in an easier way?

12. The numbers in Table 3 surprise me. To me, the category 'intervened dependency' is the most relevant since in both other categories there is not really a scarcity problem, right? Less than 2% are in this category. Oh wait, you include SSS in the 'no dependency' categoy, see my comment 9. If I include this, the number becomes 11%. This is still a low percentage in light of (my interpretation) of the literature on water scarcity. It implies that water scarcity is mostly a local problem so that not much can be expected from transboundary cooperation.

13. I find that Section 3.2 is very speculative and could perhaps be shortened.

---

## Editor Comment (EC1) · P. van der Zaag (Editor) · 14 Nov 2017

HESSD-2017-537

Editor comment on "How downstream sub-basins depend on upstream inflows to avoid scarcity: typology and global analysis of transboundary rivers" by Hafsa Ahmend Munia et al.

Pieter van der Zaag (editor)

[Figure]

The paper is of some interest to better understand the dependency of downstream sub-basin areas on upstream sub-basins. It is also quite cumbersome to read, in particular because some symbols are prone to confusion (e.g. S for stress or for scarcity or for shortage?).

I find the two reviews illuminating, critical and very constructive. I expect the authors to benefit from these comments and to significantly improve the manuscript. In so doing, some choices have to be made. The authors must clarify the added value of their typology (Fig 6) for better understanding basin trajectories

Two additional remarks that have not been made by the two reviewers:

I have one significant problem with the paper, namely that the approach is completely blue water biased and green water blind – there is no mention of green water and its importance, nor is the capacity of green water to partially substitute for blue water needs ignored. At least in the discussion section this limitation must be discussed, and the possible implications for the findings.

Related to this I have problems with the use of Falkenmark's per capita water availability as a measure of water scarcity (which the paper distinguishes from water stress). This is an old (1970s!) and very crude measure (with highly arbitrary thresholds of 1,700 m3/cap/year for stress, and 1,000 m3/cap/year for water scarcity). It was precisely Prof. Falkenmark who later introduced the very important concept of green water, which taught us that it matters a lot whether one lives in a humid (with a lot of green water) or an arid (little green water) climate, how much blue water one needs. So fixed global threshold values do make little sense.

Perhaps the paper does not need to use this flawed concept at all – omitting it may not alter the results nor the conclusions.

A second concern that was not raised is the concept of environmental water requirements / environmental flow requirements (EFRs), which are water flows that literally

run through all the SBAs and that are untouched by the riparians to safeguard the survival of aquatic ecosystems and the like. How would these feature in the typology? At least in the discussion section I would expect a reflection of the proposed method and how, if at all, EFRs could be included.

―――――――――――――――――――

---

## Author Comment (AC1) · 18 Dec 2017

**How downstream sub-basins depend on upstream inflows to avoid scarcity: typology and global analysis of transboundary rivers**

We sincerely thank the reviewers and editor for their valuable comments on this manuscript. We greatly appreciate the reviewers' input in helping to point out areas of improvement. We agree with their concerns about definitions of dependency, which have encouraged us to provide further clarity. The definitions are now updated in the revised manuscript to hopefully clarify our key conceptual innovation. To address reviewers' concern regarding the complexity of the study, we have now simplified the transition map by considering only persistent scarcity and not occasional scarcity, which gives rise to four system regimes instead of 10, connected by a simple map of transitions: NNN-SNN-SNS-SSS. This simplification of the analysis will make the typology easier to understand without compromising the key novelty of the research or the main conclusions resulting from the study. The reviewers have highlighted very useful points that helped us improve our originality by sharpening our conceptualisation, as well as description of why this approach to analysing upstream dependency is useful – tying especially to resilience literature about understanding system regime shifts. We have now substantially revised the discussion and split out a new sub-section of the discussion – 'Limitations and future work' – to address shortcomings and possible future work in more detail.

Below we first reply to the Editor's main concerns, followed by point by point response to specific questions by the editor.

**Response to Editor's comments**

**Comment 1: The paper is of some interest to better understand the dependency of downstream subbasin areas on upstream sub-basins. It is also quite cumbersome to read, in particular because some symbols are prone to confusion (e.g. S for stress or for scarcity or for shortage?). I find the two reviews illuminating, critical and very constructive. I expect the authors to benefit from these comments and to significantly improve the manuscript. In so doing, some choices have to be made. The authors must clarify the added value of their typology (Fig 6) for better understanding basin trajectories.**

> **Response 1**: We are glad to hear that editor sees the interest in this topic and we are grateful for assistance in making it easier to read.

> We have substantially revised the methodology, definitions used, as well as terminology. For the specific case of the symbol 'S', confusion arises from both 'stress' and 'shortage' being specific examples of scarcity, such that the symbol indeed stands for all three. In our revised manuscript, we have now simplified the method considerably by dropping occasional scarcity and using only scarcity (S) and no scarcity (N) to identify different scarcity categories, which has now reduced the number of variables used in the analysis.

We have now reworked the argument underlying the transition map previously shown in Figure 6 to clarify its origins and motivation. The original concept for the analysis came from the literature on resilience of socio-ecological systems, which was not sufficiently acknowledged in the previous manuscript. We regret the confusion that this omission has caused. Specifically, we use the definition from Walker et al (2004) that resilience is "*the capacity of a system to absorb disturbance and reorganize while undergoing change so as to still retain essentially the same function, structure, identity, and feedbacks*". The state of the system is defined in terms of "*state variables*", such that thresholds in those state variables are then used to define the points at which change occurs in the system function, structure, identity and feedbacks. As a short hand, we talk about a transition in system regime. Literature also talks about moving between basins of attraction and regime shifts. The former emphasizes stability and the latter tends to be associated with irreversible catastrophic failures. Our focus on transitions in system regime emphasizes simply that the system operates differently, in particular that structure, identity and feedbacks have changed (even if function may be preserved). In our revised manuscript, we are going to add this description to clearly connect the study to the resilience literature.

We agree that the original typology used in the analysis was complex and gave rise to many new terms and definitions, which were difficult to follow. The complexity arose naturally when taking into account i) both persistent and occasional scarcity and, ii) min and max of local runoff, natural discharge and actual discharge. These conditions resulted in altogether 10 system regimes (in Figure 4 of the original submission), connected by a complex map of transitions.

To reply to the reviewers' comments, we have now simplified the approach and we consider only persistent scarcity and no scarcity, leaving out occasional scarcity and using average discharge instead of min and max (as suggested by Reviewer 1). This simplification now gives rise to four system regimes (updated Figure 4), connected by a simple map of transitions: NNN-SNN-SNS-SSS (updated Figure 6) as shown below:

[Figure]

| **Scarcity under local water availability** (local runoff) | **Scarcity including upstream water** (natural discharge) | **Scarcity after accounting for upstream withdrawals** (actual discharge) | **Category name** |
|---|---|---|---|
| **No scarcity** → | No scarcity → | No scarcity ---► | NNN |
| **Scarcity** | No scarcity → | No scarcity ---► | SNN |
|  |  | Scarcity ---► | SNS |
|  | Scarcity → | Scarcity ---► | SSS |

**Scarcity and dependency category**

No dependency              N = No Scarcity
Dependency                **S = Scarcity**
Unbroken dependency
Broken dependency

*Fig 4. Definition of potential upstream water dependency categories. Dependency categories are obtained by summarizing three letter codes representing the scarcity category using runoff, natural discharge and actual discharge respectively.*

**Head water sub-basins**

[Figure]

| | | |
|---|---|---|
| N | *average.local insufficient* | S |
| *All sufficient* | | |

| | | |
|---|---|---|
| N | | S |
| | *All sufficient* | |

**Middle stream and downstream sub-basins**

[Figure]

| NNN | *average.local insufficient* | SNN | *average.afterup insufficient* | *SNS* | *average.total insufficient* | SSS |
|---|---|---|---|---|---|---|
| *All sufficient* | | | | | | |

| NNN | *All sufficient* | SNN | *average.afterup sufficient* | *SNS* | *average.total sufficient* | SSS |
|---|---|---|---|---|---|---|
* * *
*Explanation of letters*
N = No Scarcity
**S = Scarcity**

*Dependency color*
No dependency
Unbroken dependency
Broken dependency

*Dependency category under different water availability:*
N N N
└─→ Actual ('Natural' – 'Upstream water use')
└─→ Natural ('Local runoff' + 'Upstream runoff')
└─→ Local runoff

*Drivers of transitions*
Local demand: ▲ increases ▼ decreases
Upstream withdrawals: ⬤ increases ⬤ decreases

*Figure 6. Typology of groups and orders of possible transitions in dependency category, as local water demand or upstream water withdrawals (WW) increase/decrease. Upstream WWs decrease the downstream water availability, while local water demand increases the pressure on available resources given subbasin area (SBA).*

The literature on resilience and complex adaptive systems emphasizes that it is difficult to predict what will happen in future, but we can identify what are the transitions that might occur to prepare ourselves such that the system either avoids or manages those transitions. A transition map shows the system regimes and transitions for which a sub-basin may want to prepare. The new simplified version of the analysis specifically emphasizes 1) the importance of a hidden dependency, in which a subbasin may not be aware that they are avoiding scarcity because of upstream inflows, 2) the idea that the dependency may be interrupted not just due to upstream withdrawals, but also because of increases in local demand. These are fundamental ideas that are not widely recognized in existing literature.

**Comment 2: I have one significant problem with the paper, namely that the approach is completely blue water biased and green water blind – there is no mention of green water and its importance, nor is the capacity of green water to partially substitute for blue water needs ignored. At least in the discussion section this limitation must be discussed, and the possible implications for the findings.**

**Response 2:** We agree that the paper is completely blue water biased, and it was indeed a shortcoming not to mention green water at all. Specifically, we can consider the effect of green water availability

on three crucial variables in our analysis: *avail.local, avail.total, avail.afterup*. Green water availability increases the amount of locally available water by including soil water in addition to runoff. This affects scarcity, as the need for blue water should vary in response to changing green water availability, e.g. when there is less green water available, more blue water is needed. Decreases in availability of blue water (e.g. due to upstream withdrawals) may also push a region to use more green water. This is, however, a rather complex issue and not easy to quantify.

It is, however, important to note that green water is an important part of the local water availability, but by definition, it does not affect inflows from upstream. Water is called "green water" when evapotranspiration occurs directly from rain or soil water, without runoff occurring. There is no additional effect on *avail.total*, other than that on *avail.local.* Incorporating green water into our analysis will not affect our *avail.afterup* data either, as upstream withdrawals are in principle already accounted for in the water use model (including the effects of green water availability).

The thresholds for both water shortage and stress are highly uncertain, so the effect of green water on our results is difficult to anticipate. We now explicitly mention the importance of green water in the introduction and discussion, including these points.

**Comment 3: Related to this I have problems with the use of Falkenmark's per capita water availability as a measure of water scarcity (which the paper distinguishes from water stress). This is an old (1970s!) and very crude measure (with highly arbitrary thresholds of 1,700m3/cap/year for stress, and 1,000 m3/cap/year for water scarcity). It was precisely Prof. Falkenmark who later introduced the very important concept of green water, which taught us that it matters a lot whether one lives in a humid (with a lot of green water) or an arid (little green water) climate, how much blue water one needs. So fixed global threshold values do make little sense. Perhaps the paper does not need to use this flawed concept at all – omitting it may not alter the results nor the conclusions.**

> **Response 3:** We entirely agree that per capita water availability has limitations as an indicator. But we still think that both stress and shortage are useful indicators of the more general concept of scarcity. Shortage, measured by per capita water availability, captures an important intuition that sufficiency of water availability depends on population. Leaving out shortage would mean that only the stress indicator is used. This would give the impression that it is only high water use that should be avoided, not deficiency in human needs. Even though, the thresholds are arbitrary, it provides a useful balance to understand the development of water scarcity (Kummu et al. 2016), as well as illustrating the generality of our analysis framework.
>
> We already explicitly acknowledge that these are simplistic indicators, and highlight options for future work. In our new sub-section in discussion- 'Limitations and future work', we now address this issue more explicitly.

**Comment 4: A second concern that was not raised is the concept of environmental water requirements / environmental flow requirements (EFRs), which are water flows that literally run through all the SBAs and that are untouched by the riparians to safeguard the survival of aquatic ecosystems and the like. How would these feature in the typology? Atleast in the discussion section I would expect a reflection of the proposed method and how, if at all, EFRs could be included.**

> **Response 4:** We agree that EFRs are important in transboundary water management. In addition, we agree that the paper should also have explicitly mentioned environmental flow requirements. The original manuscript did mention in passing "*sustainability of water withdrawals*" but we did not elaborate the issue further. Moreover, the stress indicator includes environmental flow requirements, assuming 30% of the water is needed to satisfy the EFRs (e.g. Falkenmark et al. 2007). It is true that we do not account for EFR in a spatially disaggregated way, but global scale EFR methods could in turn be criticized for not adequately capturing on the ground conditions – our treatment of environmental flows is fit for purpose given that our focus is on the resilience-based analytical framework. In the revised manuscript, we now explicitly mention this. Further, in the new sub-section in discussion- 'Limitations and future work', we also raise this issue and suggest that the EFRs should be addressed in more detail, spatially explicitly, in possible future work on the issue.

**References**

Falkenmark, M., Berntell, A., Jägerskog, A., Lundqvist, J., Matz, M., Tropp, H. (2007). On the Verge of a New Water Scarcity: A Call for Good Governance and Human Ingenuity. SIWI Policy Brief. SIWI, Stockholm.

Kummu, M., Guillaume, J., De Moel, H., Eisner, S., Flörke, M., Porkka, M., Siebert, S., Veldkamp, T., Ward, P. J. (2016). "The World's Road to Water Scarcity: Shortage and Stress in the 20th Century and Pathways Towards Sustainability." Scientific Reports, 6, 38495.

Walker, B., Holling, C. S., Carpenter, S., Kinzig, A. (2004). "Resilience, Adaptability and Transformability in Social–ecological Systems." Ecology and Society, 9(2).

---

## Author Comment (AC2) · 18 Dec 2017

**Responses to reviewer 1- E.Mostert**

**The paper analyses the different types of dependency of downstream sub-basin areas (SBAs) on upstream SBAs, and provides a global overview of the types of dependency in 2792 SBAs. That is potentially interesting. However, the concepts used are not completely convincing and they are not always used consistently. Moreover, parts of the paper are overly complex.**

> **Response:** We sincerely thank you for your valuable comments on this manuscript. When revising the manuscript, we have considered all the comments and incorporated your suggestions to make the paper more understandable. Detailed answers to the specific questions are given below.

**Comment 1: The paper distinguishes three types of dependency of downstream SBAs on upstream SBAs: no dependency, continuous dependency, and intervened dependency. According to the definition in table 1, no dependency means that for the SBA local runoff is sufficient. Yet, in table 3 and elsewhere other cases are qualified as "no dependency" as well: SBAs that experience occasional or persistent scarcity even if they were to receive all natural runoff from upstream. That is not consistent. Moreover, in the latter cases dependency on upstream SBAs is actually high: there is already little water, and every extra drop that is used upstream results in even less water for the downstream SBA.**

**For these cases, I would introduce a fourth type of dependency, which might be called "absolute dependency"**

> **Response 1:** First, we thank the referee for pointing this out. We agree that the definitions given in the original Table 1 were not clear enough, and somewhat inconsistent with usage elsewhere in the paper. We revised the definitions and ensure in the revised manuscript that those are consistent throughout the paper. Further, we have now simplified the method of our analysis significantly to better communicate with the reader (please check our response to Editor Comment 1). The most reliable definitions are in terms of scarcity experienced with local runoff, natural discharge and actual discharge (updated Figure 4, see response to Editor comment 1 ), and the text definitions are our attempts at making these less technical.
>
> By 'No dependency', we mean that "*Upstream inflows do not influence whether or not a region experiences scarcity, i.e. if a region experiences scarcity (or not) with only local runoff, additional water from upstream does not change this situation. Note that the severity of scarcity may still be affected by upstream inflows*". Sufficiency of local runoff is a special case corresponding to the category NNN. A category SSS also implies no dependency – a sub-basin (SBA) under SSS experiences scarcity but this is not influenced by upstream inflows or water use. In other words, the sub-basin is under the same scarcity conditions regardless of upstream influence. In our revised manuscript, we updated the definitions in Table 1 and elsewhere to be more accurate and consistent. The updated definitions are:

| Term | Definition |
|------|-----------|
| *Water stress* | Demand driven water scarcity, calculated as use to availability ratio |
| *Water shortage* | Population driven water scarcity, calculated as water availability per capita |
| *Local runoff* | Runoff occurring internally within a region. |
| *Natural discharge* | Total water availability before taking into account possible upstream water withdrawals, calculated as local runoff + upstream runoff. |
| *Actual discharge* | Total water availability after upstream withdrawal; calculated as natural discharge – upstream withdrawal (local runoff + upstream runoff - upstream withdrawal). |
| *No dependency* | Upstream inflows do not influence whether or not a region experiences scarcity, i.e. if a region experiences scarcity (or not) with only local runoff, additional water from upstream does not change this situation. Note that the severity of scarcity may still be affected by upstream inflows. |
| *Dependency* | Upstream inflows influence whether a region experiences scarcity or not, i.e. how water is managed upstream can change the type of water management regime needed downstream. Two sub-types of dependency can be distinguished (as follows) |
| *Unbroken dependency* | Scarcity category is altered by upstream inflows but not by upstream water withdrawal, i.e. additional water from upstream means the region experiences no scarcity instead of scarcity. |
| *Broken dependency* | Scarcity category is altered after accounting for upstream water withdrawals, i.e. withdrawals mean that the advantages gained by upstream inflows are reduced or eliminated, and more intense water management regimes are needed downstream |

Secondly, the reviewer's description of absolute dependency refers to the severity of scarcity rather than whether or not scarcity is present or whether upstream conditions or actions influence on it. We do already acknowledge that upstream inflows can change how severe scarcity is when it occurs. While we have studied this issue previously (Munia et al. 2016), this is not the focus of the current analysis. We are instead making an argument that it is an important distinction to know whether a region would experience scarcity regardless of upstream additional inflows, or whether withdrawals might cause a scarcity category to shift. Introducing a new term (e.g. absolute dependency) would, in our opinion, make things even more complicated; we hope revising the definitions clears up the confusion.

**Comment 2: (a) Continuous dependency is defined in two slightly different ways: on p. 2 as scarcity that is avoided thanks to upstream inflow, and in table 1 as a region that would experience scarcity if it did not have access to upstream inflows. The latter formulation seems to include actual water scarcity as a result of upstream water withdrawals ("intervened dependency"). This is probably an inaccuracy.**

**(b) More problematic is that "continuous dependency" covers very different cases: cases where upstream inflow is so big that downstream scarcity is just a theoretical possibility, and cases where downstream scarcity is a serious threat because of concrete plans to increase upstream withdrawals (or plans to increase water use downstream or the effects of climate change). It would be good to distinguish between these situations, at least in the discussion.**

**(c) In addition, I would replace the term "continuous dependency" by for instance "potential dependency" because there is no dependency if scarcity is just a theoretical possibility. "Intervened dependency" could then become "actual dependency."**

**Response 2:** We reply below to each of the three points raised by reviewer:

a) The idea is to distinguish a situation where scarcity is avoided thanks to upstream inflows from a situation where scarcity does occur with upstream withdrawals, but would have been avoided with natural upstream inflows. They should therefore be mutually exclusive, but are nested in some sense. We agree that the definitions caused confusion and have now been clarified (see Reviewer 1 comment 1).

b) We thank the reviewer for this interesting observation. We have now modified the presentation of analysis to emphasise the idea that conditions with no scarcity (N) and scarcity (S) represent fundamentally different system regimes. Figure 6 and section 2.2.4 'Typology of possible transitions in dependency category' has been modified significantly to capture a simple transition of dependency from no scarcity to scarcity (see Editor comment 1).

We explicitly note that we only use simple indicators of scarcity, and encourage further work that would more rigorously investigate what it means to experience scarcity, including identifying what levels of threshold are meaningful. Future work could also quantify "distance" from a threshold, which would further address the distinction between the reviewer's two cases. The discussion section of the paper will be revised to capture the distinction between a theoretical possibility and serious threat – thank you for the suggestion.

c) The reviewer's suggestion of replacing the term "continuous dependency" by "Potential dependency" is, in our opinion, not accurate. "Potential" dependency would imply that the dependency is not currently realized. However, it is only scarcity that is not realized – scarcity is being avoided because of upstream inflows and the sub-basin therefore does not have to deal with it. The dependency is very real, it is not just a theoretical possibility – the sub-basin does need to deal with the fact that upstream withdrawals may cause them to experience scarcity. In the case of "Intervened dependency", upstream inflows no longer help – the dependency is broken – and scarcity is realized. We now propose the terms 'Broken' dependency instead of intervened dependency and 'Unbroken' dependency instead of continuous dependency.

**Comment 3: To calculate water availability in the different SBAs, the paper uses the PCR-GLOBWB model. It is not clear to me whether and how return flows were taken into account. Especially for industrial and domestic water withdrawals these can be significant.**

Response 3: In this analysis, we have used water withdrawals to calculate scarcity. Water withdrawals refer to the total amount of water withdrawn, but not necessarily consumed, by each sector; much of which is returned to the water environment where it may be available to be withdrawn again. The return flows from industrial and domestic sectors have been taken into account in PCR-GLOBWB and the recycling ratios for industrial and domestic sectors have been estimated and validated (roughly 40-80%) at a country level based on Wada et al. (2011a; 2014). We refer to Wada et al. (2011a; 2014) for the detailed descriptions. However, in this paper, estimation of return flows is uncertain and they may not necessarily be available to downstream users, for example because of pollution, timing of the flows or infiltration to groundwater (Wada et al. 2011a). Thus, the return flow was not included in the paper.

The revised method section will explicitly mention that water withdrawals provide a conservative case where return flows are not reused. The limitations section of the revised manuscript will also explicitly discuss this issue.

**Comment 4: The authors distinguish between occasional scarcity - scarcity that occurs only in a dry year - and persistent scarcity - scarcity that also occurs in a wet year. They do not define wet year and dry year. What return period is used? And why not use instead of wet year average year? Wet year water availability seems to me a very shaky basis for water scarcity management. Please reflect on this.**

Response 4: In the original submission, wet year and dry year were selected by taking into account the highest and lowest discharge occurred in 30 years (1981-2010) period respectively. To simplify the study, we are now using only scarcity and no scarcity, leaving out occasional scarcity. We use now average water availability instead of wet and dry years (please see also our response to Editor Comment 1).

However, given that the discussion paper will remain in the public record, we still wish to clarify why we included occasional scarcity. Our aim was to focus on transitions between system regimes that would require changes to either local water demand or upstream WW. Our distinction between occasional and persistent scarcity was intended to capture differences in how each type of scarcity needs to be handled. We justify the division to these two scarcity types in L222-L223 of the original manuscript, '*While persistent scarcity is obvious because of low water availability in relation to water demand, people may not necessarily be prepared for occasional scarcity, or may need adaptive measures to be actively implemented*'. It is, however, difficult to specify the conditions where adaptive vs persistent measures are needed– our definition of occasional scarcity in terms of the simple stress and shortage indicators is only indicative. Additionally, we acknowledge it is problematic that the term "occasional" applies even if only a single year has sufficient water available. We believe including occasional scarcity in future analyses is still worthwhile, especially if these limitations can be addressed.

**Comment 5: My most important concern is that the typology of possible transitions in dependency category is very complex and it is not clear to me how useful this typology is. What downstream SBAs**

**need to know is how total water availability may change as a result of climate change, how water use upstream may develop, and what their own plans and expectations are concerning water use in their own SBA. On that basis they can anticipate (an increase in) water scarcity and decide to enter into negotiations with upstream SBAs. They do not need and probably would not benefit from a full overview of groups and orders of possible transitions in dependency category.**

**Response 5:** As we mentioned in our response to Editor comment 1, it is difficult to predict what will happen in future as there is significant uncertainty around future total water availability, upstream water use and even local changes in water use. The importance of a transition pathway is that, even if we cannot anticipate the future, we can map out possible or potential transitions between system regimes that sub-basins may face, which affects both local management actions and relationships with riparian neighbours.

We have now simplified our analysis significantly (see our response to Editor Comment 1). The introduction, results and discussion are considerably revised to add more context regarding the importance of this analysis, tying to literature and terminology relating to resilience in socio-ecological systems. The typology of transitions is also simplified, while still distinguishing different experiences of scarcity and dependency as upstream withdrawals increase or decrease under each dependency type.

**Comment 6: Finally three suggestions for the presentation.**

**(a) First, the different formulation in line 255 can be simplified and made more uniform by removing "reliable" and "less reliable" and putting "dry year" and "wet year" (or "average year") always at the same place.**

**(b) Secondly, if no scarcity is N, occasional scarcity is O, why not use P for persistent scarcity?**

**(c) And thirdly, in table 4 the order in every column could be the same, e.g. always first no scarcity, then occasional scarcity, and then persistent scarcity.**

**Response 6:** Thank you for the suggestions.

a) Consistent with our reply of comment 4, we will now be using 'average year', such that the terms "reliable" and "less reliable" are no longer relevant.

b) We agree that this would have been clearer. As we dropped the occasional scarcity, in our revised manuscript we are now using only 'S' for scarcity (stress or shortage) and 'N' for no scarcity.

c) In our revised manuscript, we now focus on average years instead of wet years and dry years. As a result, Figure 4 (check Editor Comment 1) has changed significantly. We have now also arranged the columns from low to high scarcity.

**References**

Munia, H., Guillaume, J., Mirumachi, N., Porkka, M., Wada, Y., Kummu, M. (2016). "Water Stress in Global Transboundary River Basins: Significance of Upstream Water use on Downstream Stress." Environmental Research Letters, 11(1), 014002.

Wada, Y., Wisser, D., Bierkens, M. (2014). "Global Modeling of Withdrawal, Allocation and Consumptive use of Surface Water and Groundwater Resources." Earth System Dynamics Discussions, 5(1), 15-40.

Wada, Y., van Beek, L P H, Bierkens, M. F. P. (2011). "Modelling Global Water Stress of the Recent Past: On the Relative Importance of Trends in Water Demand and Climate Variability." Hydrology and Earth System Sciences, 15, 3785-3808.

Wada, Y., and Bierkens, M. F. (2014). "Sustainability of Global Water use: Past Reconstruction and Future Projections." Environmental Research Letters, 9(10), 104003.

---

## Author Response (AR1)

HESS-2017-537

**How downstream sub-basins depend on upstream inflows to avoid scarcity: typology and global analysis of transboundary rivers**

We sincerely thank the reviewers and editor for their valuable comments on this manuscript. We greatly appreciate the reviewers' input in helping to point out areas of improvement. We agree with their concerns about definitions of dependency, which have encouraged us to provide further clarity. The definitions are now updated in the revised manuscript to hopefully clarify our key conceptual innovation. To address reviewers' concern regarding the complexity of the study, we have now simplified the transition map by considering only persistent scarcity and not occasional scarcity, which gives rise to four system regimes instead of 10. This simplification of the analysis will make the typology easier to understand without compromising the key novelty of the research or the main conclusions resulting from the study. We have also changed the case study from the Dnieper to the Oder river basin after leaving out occasional scarcity.

The reviewers have highlighted very useful points that helped us improve our originality by sharpening our conceptualisation, as well as description of why this approach to analysing upstream dependency is useful – tying especially to resilience literature about understanding system regime shifts. We have now split out a new sub-section in introduction - 'A resilience perspective on upstream dependency' to provide a useful way of thinking about this problem. Discussion has been modified substantially to address shortcomings and possible future work of this analysis in more detail.

Below we first reply to the editor's main concerns, followed by point by point responses to specific questions by the reviewers.

**Response to Editor's comments**

**Comment 1: The paper is of some interest to better understand the dependency of downstream subbasin areas on upstream sub-basins. It is also quite cumbersome to read, in particular because some symbols are prone to confusion (e.g. S for stress or for scarcity or for shortage?). I find the two reviews illuminating, critical and very constructive. I expect the authors to benefit from these comments and to significantly improve the manuscript. In so doing, some choices have to be made. The authors must clarify the added value of their typology (Fig 6) for better understanding basin trajectories.**

**Response 1**:

    We are glad to hear that editor sees the interest in this topic and we are grateful for assistance in making it easier to read.

    We have substantially revised the methodology, definitions used, as well as terminology. For the specific case of the symbol 'S', confusion arises from both 'stress' and 'shortage' being specific examples of scarcity, such that the symbol indeed stands for all three. In our revised manuscript, we

have now simplified the method considerably by dropping occasional scarcity and using only scarcity (S) and no scarcity (N) to identify different scarcity categories, which has now reduced the number of variables used in the analysis.

We have now reworked the argument underlying the transition map previously shown in Figure 6 (now Figure 5) to clarify its origins and motivation. The original concept for the analysis came from the literature on resilience of socio-ecological systems, which was not sufficiently acknowledged in the original submission. We regret the confusion that this omission has caused. Specifically, we use the definition from Walker et al (2004) that resilience is "*the capacity of a system to absorb disturbance and reorganize while undergoing change so as to still retain essentially the same function, structure, identity, and feedbacks*". The state of the system is defined in terms of "*state variables*", such that thresholds in those state variables are then used to define the points at which change occurs in the system function, structure, identity and feedbacks. As a short hand, we talk about a transition in system regime. Literature also talks about moving between basins of attraction and regime shifts. The former emphasizes stability and the latter tends to be associated with irreversible catastrophic failures. Our focus on transitions in system regime emphasizes simply that the system operates differently, in particular that structure, identity and feedbacks have changed (even if function may be preserved).

In our revised manuscript, we have added this description to clearly connect the study to the resilience literature. Introduction has been updated significantly to provide the link to the resilience literature in a new sub-section (1.1) in introduction- *'A resilience perspective on upstream dependency'* -

> "*'Resilience' of a socio-ecological system is defined as "the capacity of a system to absorb disturbance and reorganize while undergoing change so as to still retain essentially the same function, structure, identity, and feedbacks" (Walker et al. 2004, p.01). Changes in the system are tracked in terms of 'state variables', such that thresholds in those state variables are used to define the points at which change occurs in the system function, structure, identity and feedbacks. When a threshold is crossed and changes occur, we say that the system has moved to a different 'basin of attraction', that there has been a 'regime shift', or a 'transition between system regimes'. While some studies aim to quantify resilience, we focus on identifying circumstances in which these regime shifts occur.*

> *Understanding thresholds and regime shifts is considered critical to adaptability and transformations in transboundary basin management (Green et al. 2013). In the case of upstream dependency, we would distinguish between different system regimes depending on whether or not water scarcity occurs and whether or not dependency occurs and its implication in the prevention of scarcity. Dependency occurs in a region when there is a transition between scarcity system regimes when considering cases where water is or is not available from upstream. We therefore compare whether scarcity occurs when water availability is calculated using solely local runoff, natural discharge (sum of local runoff and upstream runoff), and actual discharge (subtracting upstream water withdrawals from natural discharge). System regimes categorized as 'Scarcity' and 'No scarcity' are distinguished by a change in function of the system – water becomes insufficient in some sense. For the purpose of developing our analytical framework, occurrence of scarcity is determined using commonly used water shortage and water stress indicators (further discussed in Section 2.2.2). Water scarcity can also be socially induced. That is, social systems rather than climatic or hydrological factors are determining, disadvantaging groups within society, often those marginalised (Mehta 2013). Management actions may enable water to become sufficient and demonstrates a case where structural changes occur, and therefore also a transition between system regimes. However, as a first step to operationalize the concept of physical dependency over water, we focus on thresholds of physical scarcity, following existing studies (Brown and Matlock 2011, Kummu et al. 2010, Porkka et al. 2012).*

> *Transitions in system regimes in terms of dependency can occur over time, and regions can be classified according to their dependency category. Based on the role of upstream inflows and withdrawals, a region*

*might experience: i) no dependency if scarcity is not affected by upstream inflows, ii) unbroken dependency if scarcity category is altered by upstream inflows but not by upstream water withdrawal, or iii) broken dependency if scarcity is altered after accounting for upstream water withdrawals. If a system transitions into an unbroken dependency regime, the structure of the system changes – upstream withdrawals can now alter the scarcity category. An unbroken dependency is potentially hidden: a downstream part of a basin might be avoiding water scarcity only thanks to upstream inflows, and water users may not actually realise this causal factor unless those inflows are no longer available, due to increased upstream withdrawals or lower upstream runoff due to climate change or variation. That is, there is a transition to a broken dependency regime, which can also occur due to further increases in local demand. The system may then have a loss of function (insufficient water), or change in structure (due to management actions). Examining these system regimes helps to understand possible transitions of a region, and the actions that may be needed to avoid or control transition processes, e.g. negotiating water treaties to prevent or smooth transition to a broken dependency regime. We emphasise repeatedly throughout this article that upstream withdrawals may also affect the intensity of scarcity – our focus here is specifically on transitions between regimes."* (Page 2, L56-L95)

We agree with the reviewers that the original typology used in the analysis was complex and gave rise to many new terms and definitions, which were difficult to follow. The complexity arose naturally when taking into account i) both persistent and occasional scarcity and, ii) min and max of local runoff, natural discharge and actual discharge. These conditions resulted in altogether 10 system regimes (in Figure 4 of the original submission), connected by a complex map of transitions. To reply to the reviewers' comments, we have now simplified the approach and we consider only persistent scarcity and no scarcity, leaving out occasional scarcity and using average discharge instead of min and max (as suggested by *Reviewer 1*). This simplification now gives rise to four system regimes (updated Fig. 4), connected by a simple map of transitions: NNN-SNN-SNS-SSS (updated Fig. 5).

In our revised manuscript, original Figure 6 has been removed and Figure 5 has been updated to capture the changes in the transition map. Though the typology of transitions is simplified, it still distinguishes between different experiences of scarcity and dependency as upstream withdrawals increase or decrease under each dependency type. These are discussed now in more detail under discussion-sub-section (4.1) *What are the implications for mitigation and prevention of scarcity?* -

*"The literature on resilience and complex adaptive systems emphasises that it is difficult to predict what will happen in future, but we can identify what are the transitions that might occur to prepare ourselves such that the system either avoids or manages those transitions."* (Page 13, L371-L373)

And also in page 13, L385-391:

*"Understanding these transitions provides a basic level of guidance for a region. In a no dependency system regime (e.g. most SBAs analysed), efforts can be made to keep water demand at low enough levels to be self-sufficient. If water demand is expected to increase, monitoring is useful to avoid being surprised by the breaking of a hidden dependency. While our analysis shows relatively few broken or unbroken dependencies in 2010, population growth and associated water demand means that the need for water scarcity-related negotiation in transboundary basins could become a much greater issue in future. It is specifically the emergence of dependencies that introduces the need for negotiation."*

Sub-section 2.2.4 '*Typology of possible transitions in dependency category*' from the previous manuscript has been removed and significant modified text has been added in sub-section (2.2.3) – *'Determinants of dependency category and possible transitions in them'* –

*"So far, we have conceptualised change in dependency category in the context of a fixed set of water availability thresholds, obtained directly from estimated water availability volumes. The order of thresholds determines the transition in dependency category as local demand increases or decreases. In fact, even if upstream WW changes the values of the thresholds, their order will remain the same. These scarcity thresholds are naturally ordered because local water necessarily becomes insufficient before upstream water availability types respectively: local ≤ actual ≤ natural. We do, however, distinguish between headwaters vs middle stream and downstream SBAs.*

*Headwaters are the simplest case. Given they are the most upstream SBAs, they rely solely on local runoff, Increases in an SBA's demand cause transition from 'no scarcity' to 'scarcity' category. Decrease in demand would have the opposite effect (Fig 5).*

*In the case of middle stream and downstream SBAs, transition occurs between four scarcity categories, which are connected by a simple map of transitions: NNN-SNN-SNS-SSS. Transition in the scarcity category depends on both local demand and upstream WW. As the local demand increases, the SBA moves from NNN to SNN, exposing it to a 'hidden dependency' as local runoff become insufficient, but the SBA still receives sufficient upstream inflows to meet the local demand. The next transition between SNN to SNS is dependent on both local demand and upstream WW until local demand increases to the level where all available water become insufficient - the SBA becomes SSS. The decrease in local demand and upstream WW will have the opposite effect.*

*Thus SBA crosses thresholds which not only change the scarcity category, but also change the dependency category, considered in this study as transitions between different 'system regimes'. Note that we focus on the effect of increasing or decreasing local demand and upstream water withdrawal, leaving changes in water availability to future work."* (Page 9, L290-L310)

**Comment 2: I have one significant problem with the paper, namely that the approach is completely blue water biased and green water blind – there is no mention of green water and its importance, nor is the capacity of green water to partially substitute for blue water needs ignored. At least in the discussion section this limitation must be discussed, and the possible implications for the findings.**

**Response 2:**

We agree that the paper is completely blue water biased, and it was indeed a shortcoming not to mention green water at all. Specifically, we can consider the effect of green water availability on three crucial variables in our analysis: *avail.local* (local runoff)*, avail.natural* (natural discharge, i.e. including possible discharge from upstream)*, avail.actual* (actual discharge, i.e. taking into account upstream water use). Green water availability increases the amount of locally available water (*avail.local*) by including soil water in addition to runoff. This affects scarcity, as the need for blue water should vary in response to changing green water availability, e.g. when there is less green water available, more blue water is needed. Decreases in availability of blue water (e.g. due to upstream withdrawals) may also push a region to use more green water. This is, however, a rather complex issue and not easy to quantify.

It is, however, important to note that green water is an important part of the local water availability, but by definition, it does not affect inflows from upstream. Water is called "green water" when evapotranspiration occurs directly from rain or soil water, without runoff occurring. There is no additional effect on *avail.natural*, other than that on *avail.local.* Incorporating green water into our analysis will not affect our *avail.actual* data either, as upstream withdrawals are in principle already accounted for in the water use model (including the effects of green water availability).

The thresholds for both water shortage and stress are highly uncertain, so the effect of green water on our results is difficult to anticipate. We now explicitly mention the importance of green water in the discussion, including these points.

Text has now been added to the new sub-section (4.3)-*'Limitation and Future work'* –

> *"Availability of green water has not been considered either. Green water increases the amount of locally available water by including soil water in addition to runoff. This affects scarcity, as the need for blue water should vary in response to changing green water availability, e.g. when there is less green water available, more blue water is needed. Decreases in availability of blue water (e.g. due to upstream withdrawals) may also push a region to use more green water. While green water is an important part of the local water availability, it does not affect inflows from upstream, by definition. Water is called "green water" when evapotranspiration occurs directly from rain or soil water, without runoff occurring. There is no additional effect on avail.natural, other than that on avail.local. Incorporating green water into the analysis will not affect avail.actual data either, as upstream withdrawals are in principle already accounted for in the water use model (including the effects of green water availability). The thresholds for both water shortage and stress are highly uncertain, so the effect of green water on the results is difficult to anticipate."* (Page14-page15, L460-L469)

**Comment 3: Related to this I have problems with the use of Falkenmark's per capita water availability as a measure of water scarcity (which the paper distinguishes from water stress). This is an old (1970s!) and very crude measure (with highly arbitrary thresholds of 1,700m3/cap/year for stress, and 1,000 m3/cap/year for water scarcity). It was precisely Prof. Falkenmark who later introduced the very important concept of green water, which taught us that it matters a lot whether one lives in a humid (with a lot of green water) or an arid (little green water) climate, how much blue water one needs. So fixed global threshold values do make little sense. Perhaps the paper does not need to use this flawed concept at all – omitting it may not alter the results nor the conclusions.**

**Response 3:**

We entirely agree that per capita water availability has limitations as an indicator. But we still think that both stress and shortage are useful indicators of the more general concept of scarcity. Shortage, measured by per capita water availability, captures an important intuition that sufficiency of water availability depends on population. Leaving out shortage would mean that only the stress indicator is used. This would give the impression that it is only high water use that should be avoided, not deficiency in human needs. Even though, the thresholds are arbitrary, it provides a useful balance to understand the development of water scarcity (Kummu et al. 2016), as well as illustrating the generality of our analysis framework.

We already explicitly acknowledge that these are simplistic indicators, and highlight options for future work in original submission as –

*'the use of these thresholds is in line with existing studies and while there are notable limitations including that of simplification, we nonetheless utilize them as a first step in understanding upstream dependency.'* (page 8, L188-L189)  & *'Additional insights may be gained using other thresholds and/or other water scarcity indicators, such as food self-sufficiency (Gerten et al. 2011, Kummu et al. 2014) or sustainability of water withdrawals (Wada and Bierkens 2014)'*  (page 21,L491-L492).

In our revised manuscript we now address this issue more explicitly in the sub-section (2.2.2)-
*'Interpretation of upstream dependency in terms of water scarcity'*-

> *'Falkenmark's per capita water availability as a measure of water scarcity has limitations as an indicator. Nevertheless, both stress and shortage are useful indicators of the more general concept of scarcity. Shortage, measured by per capita water availability, captures an important intuition that sufficiency of water availability depends on population. Even though the thresholds are arbitrary, using both indicators provides a useful balance to understand the development of water scarcity (Kummu et al. 2016), as well as illustrating the generality of the analysis framework. The use of these thresholds is in line with existing studies and while interpretation of the results is limited by the simplicity of the indicators, they provide a first step in understanding upstream dependency.'* (Page 7, L213-L219)

**Comment 4: A second concern that was not raised is the concept of environmental water requirements / environmental flow requirements (EFRs), which are water flows that literally run through all the SBAs and that are untouched by the riparians to safeguard the survival of aquatic ecosystems and the like. How would these feature in the typology? Atleast in the discussion section I would expect a reflection of the proposed method and how, if at all, EFRs could be included.**

**Response 4:**

We agree that EFRs are important in transboundary water management. In addition, we agree that the paper should also have explicitly mentioned environmental flow requirements. The original manuscript did mention in passing "sustainability of water withdrawals" (page-21, L492) but we did not elaborate the issue further. In fact, the stress indicator does include environmental flow requirements, assuming 30% of the water is needed to satisfy EFRs (reference to Falkenmark et al. 2007 for example). It is true that we do not account for EFR in a spatially disaggregated way. In preparing this response, we did test the use of spatially variable EFRs, and note that doing so means that EFR becomes a factor that influences how stress changes between the three water availability types, i.e. it is a factor that influences dependency regime transitions. This introduces additional changes and additional complexity which makes the transition map more complex and more difficult to explain – and therefore better addressed in later publications. Moreover, global scale EFR methods could be criticized for not adequately capturing on the ground conditions – our treatment of environmental flows is fit for purpose given that our focus is on the resilience-based analytical framework.

In the revised manuscript, we now explicitly mention this in the new sub-section (4.3) in discussion-
*'Limitations and future work'*-

> *"EFRs (i.e. environmental flow requirements) are important in transboundary water management. The stress indicator used in the analysis includes EFRs, assuming 30% of the water is needed to satisfy the EFRs (reference to Falkenmark et al. 2007 for example). We do not account for EFR in a spatially disaggregated way as the analysis is conducted in the SBA scale, where spatially variable EFRs influences the dependency category, adding additional complexity to the transition map. EFRs are in any case a rather complex issue and not easy to quantify (Pastor et al. 2014). Global scale EFR methods could be criticized for not adequately capturing on the ground conditions –our treatment of environmental flows is fit for purpose given that our focus is on the resilience-based analytical framework."* (Page 14, L451-L457)

**Responses to Reviewer 1**

**The paper analyses the different types of dependency of downstream sub-basin areas (SBAs) on upstream SBAs, and provides a global overview of the types of dependency in 2792 SBAs. That is potentially interesting. However, the concepts used are not completely convincing and they are not always used consistently. Moreover, parts of the paper are overly complex.**

**Response:**

> We sincerely thank you for your valuable comments on this manuscript. When revising the manuscript, we have considered all the comments and incorporated your suggestions to make the paper more understandable. Detailed answers to the specific questions are given below.

**Comment 1: The paper distinguishes three types of dependency of downstream SBAs on upstream SBAs: no dependency, continuous dependency, and intervened dependency. According to the definition in table 1, no dependency means that for the SBA local runoff is sufficient. Yet, in table 3 and elsewhere other cases are qualified as "no dependency" as well: SBAs that experience occasional or persistent scarcity even if they were to receive all natural runoff from upstream. That is not consistent. Moreover, in the latter cases dependency on upstream SBAs is actually high: there is already little water, and every extra drop that is used upstream results in even less water for the downstream SBA.**

**For these cases, I would introduce a fourth type of dependency, which might be called "absolute dependency"**

**Response 1:**

> First, we thank the referee for pointing this out. We agree that the definitions given in the original Table 1 were not clear enough, and somewhat inconsistent with usage elsewhere in the paper. We revised the definitions and ensure in the revised manuscript that those are consistent throughout the paper. Further, we have now simplified the method of our analysis significantly to better communicate with the reader (please check our response to *Editor comment 1*). The most reliable definitions are in terms of scarcity experienced with local runoff, natural discharge and actual discharge (updated Figure 4, see response to *Editor comment 1*), and the text definitions are our attempts at making these less technical.

> By 'No dependency', we mean that "*Upstream inflows do not influence whether or not a region experiences scarcity, i.e. if a region experiences scarcity (or not) with only local runoff, additional water from upstream does not change this situation. Note that the severity of scarcity may still be affected by upstream inflows*". Sufficiency of local runoff is a special case corresponding to the category NNN. A category SSS also implies no dependency – a sub-basin (SBA) under SSS experiences scarcity but this is not influenced by upstream inflows or water use. In other words, the sub-basin is under the same scarcity conditions regardless of upstream influence.

> Secondly, the reviewer's description of absolute dependency refers to the severity of scarcity rather than whether or not scarcity is present or whether upstream conditions or actions influence it. We do already acknowledge that upstream inflows can change how severe scarcity is when it occurs. While we have studied this issue previously (Munia et al. 2016), this is not the focus of the current analysis.

We are instead making an argument that it is an important distinction to know whether a region would experience scarcity regardless of upstream additional inflows, or whether withdrawals might cause a scarcity category to shift. Introducing a new term (e.g. absolute dependency) would, in our opinion, make things even more complicated; we hope revising the definitions clears up the confusion.

In our revised manuscript, we updated the definitions in Table 1 (Page 3, L97-L98) and elsewhere to be more accurate and consistent. The updated definitions are:

| Term | Definition |
|---|---|
| *Water stress* | *Demand driven water scarcity, calculated as use to availability ratio* |
| *Water shortage* | *Population driven water scarcity, calculated as water availability per capita* |
| *Local runoff* | *Runoff occurring internally within a region (in this paper a sub-basin).* |
| *Upstream runoff* | *Runoff of the possible upstream region (in this paper a sum of runoff of upstream sub-basins)* |
| *Natural discharge* | *Total water availability before taking into account possible upstream water withdrawals, here calculated as local runoff + upstream runoff.* |
| *Actual discharge* | *Total water availability after upstream water withdrawals; calculated as natural discharge – upstream withdrawals (local runoff + upstream runoff – upstream withdrawals).* |
| *No dependency* | *Upstream inflows do not influence whether or not a region experiences scarcity, i.e. if a region experiences scarcity or not with only local runoff, additional water from upstream does not change this situation. Note that the severity of scarcity may still be affected by upstream inflows and water withdrawals.* |
| *Dependency* | *Upstream inflows influence whether a region experiences scarcity or not, i.e. how water is managed upstream can change the type of water management regime needed downstream. Two sub-types of dependency can be distinguished (as follows).* |
| *Unbroken dependency* | *Scarcity category is altered by upstream inflows but not by upstream water withdrawals, i.e. additional water from upstream means the region experiences no scarcity instead of scarcity and upstream withdrawals do not change this.* |
| *Broken dependency* | *Scarcity category is altered after accounting for upstream water withdrawals, i.e. withdrawals mean that the advantages gained by upstream inflows are reduced or eliminated, and more intense water management regimes are needed downstream.* |

**Comment 2: (a) Continuous dependency is defined in two slightly different ways: on p. 2 as scarcity that is avoided thanks to upstream inflow, and in table 1 as a region that would experience scarcity if it did not have access to upstream inflows. The latter formulation seems to include actual water scarcity as a result of upstream water withdrawals ("intervened dependency"). This is probably an inaccuracy.**

**(b) More problematic is that "continuous dependency" covers very different cases: cases where upstream inflow is so big that downstream scarcity is just a theoretical possibility, and cases where downstream scarcity is a serious threat because of concrete plans to increase upstream withdrawals (or plans to increase water use downstream or the effects of climate change). It would be good to distinguish between these situations, at least in the discussion.**

**(c) In addition, I would replace the term "continuous dependency" by for instance "potential dependency" because there is no dependency if scarcity is just a theoretical possibility. "Intervened dependency" could then become "actual dependency."**

**Response 2:** We reply below to each of the three points raised by reviewer:

a)  The idea is to distinguish a situation where scarcity is avoided thanks to upstream inflows from a situation where scarcity does occur with upstream withdrawals, but would have been avoided with natural upstream inflows. They should therefore be mutually exclusive, but are nested in some sense.

    We agree that the definitions caused confusion and have now been clarified. (also see our responses to *Reviewer 1 comment 1*).

b)  We thank the reviewer for this interesting observation. We have now modified the presentation of analysis to emphasise the idea that conditions with no scarcity (N) and scarcity (S) represent fundamentally different system regimes. We explicitly note that we only use simple indicators of scarcity, and encourage further work that would more rigorously investigate what it means to experience scarcity, including identifying what levels of threshold are meaningful. Future work could also quantify "distance" from a threshold, which would further address the distinction between the reviewer's two cases as we now mentioned in the text -

    *"Future work could also quantify "distance" from a threshold, which would further address the distinction between how close these basins are to scarcity."* (Page 15, L481-L482)

    The discussion section of the paper is now revised to capture these limitations under sub-section (4.3) *Limitations and future work*– thank you for the suggestion.

c)  The reviewer's suggestion of replacing the term "continuous dependency" by "Potential dependency" is, in our opinion, not accurate. "Potential" dependency would imply that the dependency is not currently realized. However, it is only scarcity that is not realized – scarcity is being avoided because of upstream inflows and the sub-basin therefore does not have to deal with it. The dependency is very real, it is not just a theoretical possibility – the sub-basin does need to deal with the fact that upstream withdrawals may cause them to experience scarcity. In the case of "Intervened dependency", upstream inflows no longer help – the dependency is broken – and scarcity is realized. We have now used the terms 'Broken' dependency instead of intervened dependency and 'Unbroken' dependency instead of continuous dependency in our revised manuscript.

**Comment 3: To calculate water availability in the different SBAs, the paper uses the PCR-GLOBWB model. It is not clear to me whether and how return flows were taken into account. Especially for industrial and domestic water withdrawals these can be significant.**

**Response 3:**

In this analysis, we have used water withdrawals to calculate scarcity. Water withdrawals refer to the total amount of water withdrawn, but not necessarily consumed, by each sector; much of which is returned to the water environment where it may be available to be withdrawn again. The return flows from industrial and domestic sectors have been taken into account in PCR-GLOBWB and the recycling ratios for industrial and domestic sectors have been estimated (roughly 40-80%) and validated at a country level based on Wada et al. (2011a, 2014). We refer to Wada et al. (2011a, 2014) for the detailed descriptions. However, in this paper, estimation of return flows is uncertain and they may not necessarily be available to downstream users, for example because of pollution, timing of the flows or infiltration to groundwater (Wada et al. 2011a). Thus, the return flow was not subtracted from withdrawals in the paper.

The revised *Data* section (2.1) is now explicitly mentioning this-

*"The return flows from industrial and domestic sectors have been taken into account in PCR-GLOBWB and the recycling ratios for industrial and domestic sectors have been estimated (roughly 40-80%) at a country level and validated based on Wada et al. (2011a, 2014)."* (Page 4, L133-L135)

*The 'Limitations and future work section' (4.3) of the revised manuscript also explicitly discuss this issue.*

*"we used water withdrawals, which refer to the total amount of water withdrawn, but not necessarily consumed, by each sector; much of which is returned to the water environment where it may be available to be withdrawn again. The return flows from industrial and domestic sectors have been taken into account in PCR-GLOBWB and the recycling ratios for industrial and domestic sectors have been estimated and validated (roughly 40-80%) at a country level based on Wada et al. (2011a, 2014). However, in this paper, estimation of return flows is uncertain and they may not necessarily be available to downstream users, for example because of pollution, timing of the flows or infiltration to groundwater (Wada et al. 2011a). We therefore did not include return flows when calculating water stress, but could in future."*(Page-14, L443-L450)

**Comment 4: The authors distinguish between occasional scarcity - scarcity that occurs only in a dry year - and persistent scarcity - scarcity that also occurs in a wet year. They do not define wet year and dry year. What return period is used? And why not use instead of wet year average year? Wet year water availability seems to me a very shaky basis for water scarcity management. Please reflect on this.**

**Response 4:**

In the original submission, wet year and dry year were selected by taking into account the highest and lowest discharge occurred in 30 years (1981-2010) period respectively. To simplify the study, we are now using only scarcity and no scarcity, leaving out occasional scarcity. We now use average water availability instead of wet and dry years (please see also our response to *Editor Comment 1*).

However, given that the discussion paper will remain in the public record, we still wish to clarify why we included occasional scarcity. Our aim was to focus on transitions between system regimes that

would require changes to either local water demand or upstream water withdrawals (WW). Our distinction between occasional and persistent scarcity was intended to capture differences in how each type of scarcity needs to be handled. We justify the division to these two scarcity types in L222-L223 of the original manuscript, '*While persistent scarcity is obvious because of low water availability in relation to water demand, people may not necessarily be prepared for occasional scarcity, or may need adaptive measures to be actively implemented*'. It is, however, difficult to specify the conditions where adaptive vs persistent measures are needed– our definition of occasional scarcity in terms of the simple stress and shortage indicators is only indicative. Additionally, we acknowledge it is problematic that the term "occasional" applies even if only a single year has sufficient water available. We believe including occasional scarcity in future analyses is still worthwhile, especially if these limitations can be addressed.

**Comment 5: My most important concern is that the typology of possible transitions in dependency category is very complex and it is not clear to me how useful this typology is. What downstream SBAs need to know is how total water availability may change as a result of climate change, how water use upstream may develop, and what their own plans and expectations are concerning water use in their own SBA. On that basis they can anticipate (an increase in) water scarcity and decide to enter into negotiations with upstream SBAs. They do not need and probably would not benefit from a full overview of groups and orders of possible transitions in dependency category.**

**Response 5:**

As we mentioned in our response to *Editor comment 1*, it is difficult to predict what will happen in future as there is significant uncertainty around future total water availability, upstream water use and even local changes in water use. The importance of a transition pathway is that, even if we cannot anticipate the future, we can map out possible or potential transitions between system regimes that sub-basins may face, which affects both local management actions and relationships with riparian neighbours. We have now simplified our analysis significantly (see our response to *Editor Comment 1*).

The introduction, results and discussion are considerably revised to add more context regarding the importance of this analysis, tying to literature and terminology relating to resilience in socio-ecological systems. The typology of transitions is also simplified, while still distinguishing different experiences of scarcity and dependency as upstream withdrawals increase or decrease under each dependency type (see our response to *Editor Comment 1*).

**Comment 6: Finally three suggestions for the presentation.**

(a) **First, the different formulation in line 255 can be simplified and made more uniform by removing "reliable" and "less reliable" and putting "dry year" and "wet year" (or "average year") always at the same place.**

(b) **Secondly, if no scarcity is N, occasional scarcity is O, why not use P for persistent scarcity?**

(c) **And thirdly, in table 4 the order in every column could be the same, e.g. always first no scarcity, then occasional scarcity, and then persistent scarcity.**

**Response 6:** Thank you for the suggestions.

a) Consistent with our reply of *Reviewer 1 comment 4*, we will now be using 'average year', such that the terms "reliable" and "less reliable" are no longer relevant.

b) We agree that this would have been clearer. As we dropped the occasional scarcity, in our revised manuscript we are now using only 'S' for scarcity (stress or shortage) and 'N' for no scarcity.

c) In our revised manuscript, we now focus on average years instead of wet years and dry years. As a result, Figure 4 (check *Editor Comment 1*) has changed significantly. We have now also arranged the columns from low to high scarcity.

**Responses to Reviewer 2**

**This paper is on the interconnectedness of water withdrawals and water scarcity in transboundary basins. A method is presented to formally analyze this interconnectedness. This method is subsequently applied to a global hydrological model. The results show that (in my interpretation), interconnectedness is generally low. The implication is that water scarcity is mostly a local problem, which is new to me. My overall assessment is that this paper is a solid piece of work with a new result that has changed my perspective on the management of transboundary rivers, and I thank the authors for this contribution. I do have some comments. Most of them relate to a lack of precision in the use and application of definitions and terminology. My comments may have implications for (the presentation of) both analysis and results.**

> **Response:** We sincerely thank you for your supportive words and constructive comments. We have taken all your comments and suggestions into account when revising the paper. Detailed answers to the specific comments are given below.

**Major Comments**

**Comment 1 (a): The definitions of dependency in Table 1 are not mutually exclusive, although they should be. A visual representation of the authors' definitions and my proposal to adjust them are displayed in Figure 1 below. Adjustment would probably have some consequences for the analysis, which I hope/expect are easy to incorporate. If not, one simplification would be to merge the 'dep' and 'oops' categories. Perhaps the resulting categorization is the one intended by the authors. An even simpler, and perhaps more relevant categorization is to not only merge 'dep' and 'oops', but also merge 'no dep' and 'still no dep'. Results and insights will stay the same but the presentation will be easier.**

[Figure]

Figure 1: Dependency. Top plot according to Table 1. Bottom plot proposed by reviewer.

**Response 1:** We thank the referee for pointing this out, and showing that we need to further clarify our definitions. In the revised submission, the definition of dependency is now (described in updated Table 1; see reply to *Reviewer 1 comment 1*)*: ''Upstream inflows influence whether a region experiences scarcity or not, i.e. how water is managed upstream can change the type of water management regime needed downstream.''* This means, upstream water dependency occurs if water from upstream is needed to avoid scarcity and by scarcity category (NNN, SNN, SNS, SSS) we mean (described in updated Fig 4; see response to *Editor comment 1*) ''the *stress or shortage condition of a sub-basin under different water availability (local, natural & actual)*''.

Our definition of "No dependency" now states "*Upstream inflows do not influence whether or not a region experiences scarcity, i.e. if a region experiences scarcity (or not) with only local runoff, additional water from upstream does not change this situation*". With our simplified analysis, this includes only two cases: NNN and SSS. The first experiences no scarcity regardless of whether upstream inflows are available. The second does experience scarcity, again, regardless of whether upstream inflows are available.

As a result, the graphical representation of Table 1 by *reviewer 2* does not quite reflect the key distinctions in our analysis, as the 'no dependency' condition can happen both with and without scarcity. Sufficiency of local runoff is a special case corresponding to the category NNN. Exceeding local runoff does not necessarily mean a region is dependent.

We note that the definitions in Table 1 were not clear enough in the original submission, and inconsistent with usage elsewhere in the paper. We have rectified this confusion.

In our revised manuscript, the definitions in Table 1 and elsewhere are updated to be accurate and consistent with the analysis. The updated definitions of dependency are presented in response to *Reviewer 1 comment 1*. At the same time, we have now simplified our analysis taken into account only scarcity and no scarcity, which reduces the number of definitions and categories as well as simplifying the transition map, as presented in response to *Editor comment 1*.

**Comment 2: The terms used in Table 1 are not used consistently in the text.**

**a) The authors use the terms runoff and discharge interchangeably (even in Table 1), which is confusing.**

> **Response 2a:** We used both the terms runoff and discharge depending on context. By runoff, we mean that part of the precipitation, snowmelt, or irrigation water that appears in surface streams, while discharge refers to flow (accounting for routing of runoff). We now use the term 'local runoff' for this. As noted in the paper, we approximate discharge as the sum of local runoff in local and upstream sub-basins, such that there is an arithmetic relationship between the two. We used the local runoff data for every sub-basin area (SBA) to calculate their own water availability, while in natural discharge the local runoff from each upstream SBA was added to the SBA's local runoff. Actual discharge was calculated from this by subtracting the water withdrawals in upstream SBAs.

> With respect to Table 1 specifically, local runoff is not discharge because it excludes upstream inflows, upstream inflows are the sum of upstream runoff, so the two terms are indeed interchangeable, natural discharge is calculated as the sum of local and upstream runoff, so is defined as such. Actual discharge is local + upstream runoff - upstream withdrawals, but is more easily defined by comparison to natural discharge.

> This is now better explained in the revised manuscript, with explicit definitions in Table 1.

**b) The terms 'water withdrawals', 'need' and 'demand' are introduced as different concepts (L129) but they lack proper definitions. Perhaps 'demand' should be replaced by 'quantity demanded', which is something different, or 'use'.**

> **Response 2b:** Thanks – we agree these concepts needed clarification. These terms are used in three different contexts within the paper, for calculation of water availability after upstream withdrawals, water stress, and water shortage. Water withdrawal is water withdrawn from a surface water or groundwater source for domestic, industrial and agricultural use. Calculation of water stress uses water withdrawal data to reflect impacts from high use of water. Calculation of water shortage focuses on need for water, in terms of per capita water availability. "Demand" was used as a high-level umbrella term covering both actual withdrawals and need for water (as understood by the water shortage indicator). We believe "quantity demanded" would be too specific in this case (and more cumbersome), and "use" would not cover the idea of "need".

> Explicit definitions have now been given in the sub-section (2.2.3) - *Determinants of dependency category and possible transitions in them* –

> > *"In this study, 'demand' is used as a high-level umbrella term covering both actual withdrawals (for the stress indicator) and need for water (population, for the shortage indicator*)." (Page 9, L274-L275)

**c) In L164 the term 'discharge after upstream WW' is used where authors probably refer to 'actual discharge' from Table 1. In the same paragraph, variable 'avail.afterup' is 1st introduced to reflect the same term, so that we now have three terms for the same concept. More variables are then introduced that face the same problem. This is really confusing and obscures the line of argumentation in the main text.**

**Response 2c**: We agree with the reviewer regarding this issue. These three terms (discharge after upstream WW, actual discharge, *avail.afterup*) referred in the original submission to the same type of discharge. We used *avail.afterup* as the short form to fit in the transition map. So, *avail.afterup* can be consider as the symbolic representation of actual discharge. 'Discharge after upstream WW', in turn, was used to explain what 'actual discharge' means.

In the revised manuscript, we are using *avail.actual* instead of *avail.afterup* and we make sure that this term is explained only once at the beginning and we use the term consistently for the rest of the manuscript.

**Comment 3(a) another comment on Table 1. The order of presentation is illogical and should be reversed. Start with water stress/shortage, which you need to understand scarcity, then runoff/discharge, both of which you need to understand dependency.**

**Response 3(a):** Thank you for the suggestion. We have revised the order of presentation in Table 1 accordingly.

**Comment 3 (b): A more bold suggestion is the following. Since you assign variable names to some of the terms in Table 1, it would perhaps be transparent to introduce a formula for dependency (with shorter variable names), which would make it much easier to understand the definitions. For example, if qi denotes water use in sub-basin i, ei denotes local runoff, and Pi denotes the set of i's predecessors (i.e. sub-basins strictly upstream of i) we can write:**

- $\hat{e}_i = e_i + \sum_{j \in P_i}(e_j)$ as the total water available after upstream withdrawal;
- $\bar{e}_i = e_i + \sum_{j \in P_i}(e_j - q_j)$ as the total water available after upstream withdrawal.

Subsequently, when we denote $x_i$ as the measure of water needed to avoid water scarcity (be it from stress or shortage), we have:

- $x_i \leq e_i \quad \rightarrow \quad$ no dependency;
- $e_i < x_i \leq \hat{e}_i \quad \rightarrow \quad$ still no dependency (see bottom plot of Figure 1);
- $\hat{e}_i < x_i \leq \bar{e}_i \quad \rightarrow \quad$ dependency.

**These formalizations of the definitions may also assist in discussing e.g. the typology of dependence categories in Section 2.2.4. I realize that I might be pushing this point too far. If this is the case then at least sharpen and streamline the definitions and terms used in the paper in a consistent way.**

**Response 3(b):** As we mentioned in our response to *Reviewer 2 comment 1*, upstream water dependency occurs if upstream inflows influence whether a region experiences scarcity or not. Currently, the definitions in the manuscript are creating confusion and as also mentioned in our reply to *Reviewer 2 comment 1,* that dependency is not captured by the diagram suggested by the reviewer, such that the suggested variables would likely be insufficient.

Even though we agree that the symbols would provide shorter variable names, use of symbols would increase the level of abstraction and might make it more difficult to understand. We have now modified the definitions of dependencies, which we believe will further clarify the concept.

**Comment 4: While you mention treaties on transboundary river water in the discussion, they seem to be ignored in the analysis. Dependencies may not be as severe when they are mitigated by treaties that provide security of continuous upstream inflow. Such treaties may even feature well-designed (flexible) sharing rules able to mitigate the impacts of e.g. climate change. We could even have reversed dependency when a treaty stipulates that local runoff should be shared with downstream riparians. In this case, even if local runoff would be sufficient to satisfy demand, the upstream country would be dependent on the downstream country(/-ies). An example would be Ethiopia's position in the Blue Nile basin.**

**Response 4:** The main focus of our analysis is to identify physical upstream water dependency and explain its direct drivers. The effectiveness of possible treaties was not analysed; instead, our aim was to briefly discuss how this analysis could help treaties to better address water scarcity problems. Treaties have only an indirect effect on physical upstream water dependency, as we define it. They affect development of water resources locally and upstream, which may change the dependency status. It is not the dependency status that would be less severe with a treaty rather than without one, rather a well-designed treaty would attempt to provide interventions that influence the stability of the dependency and hence prevent scarcity from occurring.

The idea of reverse dependencies would be interesting to pursue in future work. Rather than taking a purely physical view of the river system, we can consider a binding downstream allocation as a form of water use that reduces availability, in similar terms to upstream withdrawals. At the same time, the allocation can be considered to increase downstream local availability – the water might be considered to have an equivalent status to local runoff.

Discussion has been revised to reflect these links with treaties under section 4.1 *'What are the implications for mitigation and prevention of scarcity?'* –

> *"While our analysis shows relatively few broken or unbroken dependencies in 2010, population growth and associated water demand means that the need for water scarcity-related negotiation in transboundary basins could become a much greater issue in future. It is specifically the emergence of dependencies that introduces the need for negotiation. Treaties have an indirect effect on physical upstream water dependency by limiting or coordinating development of water resources locally and upstream. Treaty design can be innovated to include functions that improve the stability of the dependency and hence prevent scarcity from occurring. If decision makers cannot avoid a transition to scarcity (i.e. a broken dependency), perhaps due to factors outside their control, then coordination can at least facilitate adaptation to cope with physical water scarcity. There are regions where physical water scarcity is to some extent expected – development is limited by water availability, such that fully utilising other resources (e.g. land) requires more water than is available. In addition, it should be pointed out that negotiation for rights to upstream inflows is only one strategy among many to try to meet water demand. In such cases, treaties can focus on mitigating the severity of impacts of scarcity."*(Page-13, L388-L399).

**Minor Comments:**

**Comment 5 L53: Please define 'sub-basin' upon first use.**

**Response 5:** The manuscript is corrected.

**Comment 6. L53: 'experiences' → 'may experience'.**

**Response 6:** The manuscript is corrected.

**Comment 7. L55: 'Parts of basins' do not 'realise' much.**

**Response 7:** Manuscript has been updated.

Sentence in the original manuscript: *"We argue that a sub-basin therefore experiences a 'hidden' dependency: a downstream part of a basin might be avoiding water scarcity only thanks to upstream inflows, and may not actually realize it until those inflows are no longer available due to increased upstream withdrawals or lower runoff due to potential climate change impacts."*

We have now reworded the sentence to provide a stronger link with the dependency regime and avoid implying presence of an actor. Sentence is now revised as follows:

> *"If a system transitions into an unbroken dependency regime, the structure of the system changes – upstream withdrawals can now alter the scarcity category. An unbroken dependency is potentially hidden: a downstream part of a basin might be avoiding water scarcity only thanks to upstream inflows, and water users may not actually realise this causal factor unless those inflows are no longer available, due to increased upstream withdrawals or lower upstream runoff due to climate change or variation."* (Page-2, L85-L89)

**Comment 8. Figure 1 duplicates Table 1 and can be removed.**

**Response 8:** Table 1 provides definitions, whereas Figure 1 provides a graphical overview of contributions of the paper (similar to a graphical abstract). We would prefer to keep both, particularly to address different learning styles of the reader.

**Comment 9. L131: The 30yr period is introduced here without any explanation. Why? And how?**

**Response 9:** The 30yr period was used to capture the current hydro climatic characteristics, with water availability calculated as summary values (in the new version, average water availability).

Explanation added in the sub section 2.2.2-*Interpretation of upstream dependency in terms of water scarcity –*

> *'The 30-year period was used to capture the current hydro climatic characteristics.'* (Page 7, L227)

**Comment 10. L155–159: Are return flows accounted for?**

**Response 10:** In our analysis, return flows are assumed not to be usable downstream. Withdrawal refers to the total amount of water used for each sector, much of which is returned to the water environment where it may be available to be withdrawn again. However, estimation of return flows is uncertain and they may not necessarily be available to downstream users, for example because of pollution, timing of the flows or infiltration to groundwater (Wada et al. 2011a, Wada et al. 2011b). Thus, the return flows were not included in the paper.

The revised method section explicitly mentions that our analysis provides an extreme case where return flows are not reused (see also Munia et al. 2016). The limitations section (4.3) of the revised manuscript also explicitly discusses this issue (Page-14, L445-*L450) (Please see our response to Reviewer 1 comment 3)*

**Comment 11. L165–166: What if an SBA has multiple downstream SBAs? Possibility of double-counting.**

**Response 11:** The sentence in the original manuscript reads as follows: "*We identified the entire upstream area for each SBA based on the upstream-downstream hierarchy; i.e. in cases when an SBA has more than one upstream SBA, the total upstream water use is summed (WW.upstream)."*

The drainage network used here to identify upstream-downstream has a clear hierarchical relation, with no distributaries, so water only flows to one immediately downstream sub-basin and there is no risk of double counting. This is now explicitly said on page 6, L185.

**Comment 12. L198–199: What happened to 'persistent' and 'occasional' from Table 1?**

**Response 12:** Occasional scarcity has now been dropped from the analysis (see response to *Editor comment 1*).

**Comment 13. Figure 4: the color code categorizes SSS as featuring 'no dependency' which seems incorrect. In general, I would say that any setting where there is scarcity under actual discharge (i.e. after upstream water use) should be coded as 'intervened dependency', since the upstream water use exacerbates the downstream scarcity. I realize that the authors would probably say that this is a case of 'no dependency' because there would also be scarcity with out upstream water use, but that is a semantic argument since scarcity is coded here as a binary variable.**

**Response 13:** Thank you for sharing your interpretation. Please see our response to *Reviewer 1 comment 1*. We have updated the definitions in Table 1. All upstream water use affects the severity (or frequency) of downstream scarcity, so all situations would be coded as intervened dependency by that definition, reducing its utility. In our case, we are more interested in the transition between discrete system regimes (also see response to *Editor Comment 1*), which is why we have not adopted the reviewer's suggestion.

**Comment 14. The term 'ordering' and the arrows used in Figures 6 and 8 suggest that sub-basins can only develop in one direction, namely from good (NNN) to bad (SSS). You may want to present a more nuanced story, explaining under what circumstances this tendency may be reversed.**

> **Response 14:** Thanks for raising this issue. Indeed, we do not want to give the impression that SBAs can only develop in one direction; we do already mention the possibility of changes in the other direction at several points.
>
> The change in dependency category goes backward with the decrease of own and upstream water withdrawal as explained in L280 of original manuscript: "*Over time, this change in dependency category could go forward and backward as water demand of the SBA increases or decreases*" (Page 9, L286-L288, in revised manuscript) and in L295: "*decrease in demand would have the opposite effect*" (page 9,L297 in revised manuscript).
>
> We have now modified Figure 5 (see *Editor Comment 1*, updated Figure 5) and added text to sub-section (2.2.3) *Determinants of dependency category and possible transitions in them* -to better explain the reversed condition (see *Editor Comment 1)*.

**Comment 15. Figure 6 is presenting too much at the same time. From the text I understand that there is a natural ordering, but I do not see the added value of presenting all possible pathways through these orders. Same of course for Figure 8. Can you somehow summarize this in an easier way?**

> **Response 15:** As noted in response to *Reviewer 1 comment 5 & Editor Comment 1*, we agree that the original typology of transitions was complex, but this was what emerged from our simple set of assumptions when trying to map out system regimes and potential transitions between them. We have now simplified the analysis by taking into account only scarcity and no scarcity conditions (i.e. not considering occasional scarcity) and using average discharge instead of min and max discharges. This simplification now results in four system regimes (see updated Figure 4 and *Editor Comment 1*), connected by a one simple map of transitions. We also further motivated within the paper why we are interested in looking at a transition map in the first place by connecting the study with the concept of regime shifts, from the resilience literature (see also *Editor comment 1*).

**Comment 16. The numbers in Table 3 surprise me. To me, the category 'intervened dependency' is the most relevant since in both other categories there is not really a scarcity problem, right? Less than 2% are in this category. Oh wait, you include SSS in the 'no dependency' categoy, see my comment 13. If I include this, the number becomes 11%. This is still a low percentage in light of (my interpretation) of the literature on water scarcity. It implies that water scarcity is mostly a local problem so that not much can be expected from transboundary cooperation.**

> **Response 16:** It's important to distinguish between dependency and scarcity and recognize that dependency is primarily about potential for future scarcity, which transboundary cooperation aims to mitigate. The "intervened dependency" (in current manuscript 'broken dependency') category and SSS only include cases where institutional arrangements have failed to prevent scarcity from occurring – that it is a low percentage is reassuring, because it suggests that transboundary cooperation has not

too frequently failed. It is, however, debatable whether 11% is a low percentage from that point of view.

To judge the importance of transboundary cooperation, it is more important to look at areas with no scarcity who are dependent on upstream inflows. New results (excluding occasional scarcity) in the abstract highlights that "*386 million people (14%) live in SBAs that can avoid stress owing to available water from upstream and have thus upstream dependency. In the case of water shortage, 306 million people (11%) live in SBAs dependent on upstream water to avoid possible shortage.*"

While these percentages are similar to those cited by the reviewer (and therefore arguably low), it is important to note that the results look very different after considering occasional scarcity, e.g. in drought conditions. The previous manuscript (including occasional scarcity) stated in the abstract that "*Our results show that almost 932 million people (33% of the total transboundary population) live in SBAs that are dependent on upstream water to avoid stress because of their own water use, while 464 million people (17% of the total transboundary population) live in SBAs dependent on upstream water to avoid possible shortage*". While our analysis does not consider how close these basins are to scarcity (as pointed out by *Reviewer 1 comment 2*), it is clear that transboundary cooperation is widely important for avoiding deterioration of the current status quo – it is not just a local problem. We do, however, agree that it is more of a local problem than the literature often recognizes. We also have now revised the discussion to reflect the above-mentioned implications of our results for transboundary cooperation under section 4.2 *Relation to existing work*- as mentioned now in Page 14, L437-L441-

> *"our analysis highlights the importance of local demand in causing scarcity and dependency. If local demand stays low enough and local water resources are sufficient to meet the demand, neither scarcity nor dependency occurs, and transboundary cooperation is not needed. This point has been made in existing literature (e.g. related to social construction of scarcity). These are fundamental ideas that are not widely recognized in existing literature."*

And also in Page-13-14, L412-L418,

> *"Our work distinguishes between dependency and scarcity and recognizes that dependency is primarily about potential for future scarcity, which transboundary cooperation aims to mitigate. To judge the importance of transboundary cooperation, it is more important to look at areas under no scarcity which are dependent on upstream inflows. The "broken dependency" category (SNS) and SSS only include cases where institutional arrangements have failed to prevent scarcity from occurring. Our work, however, highlights that negotiation to avoid needing to cope with scarcity is only part of the issue. As demand increases, negotiation among riparian countries will eventually turn to discussion of intensity and frequency of scarcity, and the level of demand at which it occurs."*

**Comment 17. I find that Section 3.2 is very speculative and could perhaps be shortened.**

**Response 17:** Section 3.2 has been removed because of occasional scarcity has been left out from the analysis – the full typology no longer needs to be described.

For the record, we note that the old Section 3.2 was already acknowledged in the manuscript as speculative, with the aim of providing possible implications of the ordering and most importantly its connection with water demand and water availability. The section tried to make suggestions for how

negotiation in upstream-downstream relationships might be influenced, which was important to understand the significance of the transition maps identified in the analysis.

**References**

[revised manuscript text omitted]

Dnieper basin case study

| Sub basin areas (SBAs) | Rivers | Water Availability [$10^9$ m$^3$ yr$^{-1}$] |
|---|---|---|

**Sub basin areas (SBAs)**
- Upstream
- Middle stream
- Downstream

**Rivers**
- River network
- Flow direction at country border

**Water Availability [$10^9$ m$^3$ yr$^{-1}$]**
- Local runoff
- Natural ('Local runoff' + 'Upstream runoff')
- Actual ('Natural' – 'Upstream water use')

DnSBA$_{RU\text{-}A}$

*basin*   *country*   *country sub-code*

[Figure]

*Fig. 2. Upstream-downstream relationship between* *SBAs) in the* *Oder basin and average simulated*  *water availability for 1981-2010. Drainage network and sub-basin division are based on DDM30* ~~(Döll 2002) and country borders (Natural Earth 2017) with additional manual assignment of border cells.*

**Interpretation of upstream dependency in terms of water scarcity**

Looking at the average availability of water (1981-2010) for the SBAs of the Oder basin provides an illustration of the concept of upstream dependency (Fig 2). The headwater SBA (OdSBA$_{CZ}$) obviously has no upstream dependency; the three types of water availability are the same. But in the case of SBAs OdSBA$_{PO-A}$, OdSBA$_{PO-B}$, OdSBA$_{GE}$ upstream water availability and withdrawals influence water availability. These are the SBAs we are most interested in.

ependency on upstream water can  be assessed by comparing an SBA's scarcity category across the different water availability types (i.e. local runoff, natural discharge, actual discharge – see definitions in Table 1). We calculated scarcity using  water stress and water shortage indices. Water stress refers to impacts from high use of water while water shortage refers to impacts from insufficient water availability per person (Falkenmark et al. 2007, Kummu et al. 2016).

The stress indicator was calculated as WW.local/avail and the shortage indicator is calculated as *avail/population.local*. The stress indicator includes environmental flow requirements (EFRs), assuming 30% of the water is needed to satisfy the EFRs (Falkenmark et al. 2007). To determine whether water

stress or shortage occurs, we respectively used the thresholds 0.2 and 1000 m³/capita/yr, as defined by Falkenmark et al (2007) and used by other research too (Liu et al. 2017). Crossing this threshold leads to impacts from insufficient water availability per person, potentially limiting economic development, and human health and well-being (Falkenmark et al. 2007). Falkenmark's per capita water availability as a measure of water scarcity has limitations as an indicator. Nevertheless, both stress and shortage are useful indicators of the more general concept of scarcity. Shortage, measured by per capita water availability, captures an important intuition that sufficiency of water availability depends on population. Even though the thresholds are arbitrary, using both indicators provides a useful balance to understand the development of water scarcity (Kummu et al. 2016), as well as illustrating the generality of the analysis framework. The use of these thresholds is in line with existing studies and while interpretation of the results is limited by the simplicity of the indicators, they provide a first step in understanding upstream dependency.

Eqwith 1) local runoff, 2) natural discharge and 3) actual discharge. Equations for water stress:

1. $\dfrac{\textit{withdrawal.local}}{\textit{avail.local}}$ ; 2. $\dfrac{\textit{withdrawal.local}}{\textit{avail.total}}$ ; 3. $\dfrac{\textit{withdrawal.local}}{\textit{avail.total} - \textit{withdrawal.upstream}}$ $\dfrac{WW.local}{avail.local}$ ; 2. $\dfrac{WW.local}{avail.natural}$ ;
3. $\dfrac{WW.local}{avail.natural - WW.upstream}$ .

Equations for water shortage:

1. $\dfrac{avail.local}{population.local}$ ; 2. $\dfrac{\textit{avail.total}}{\textit{population.local}}$ $\dfrac{avail.natural}{population.local}$ ; 3. $\dfrac{avail.\textit{total} - \textit{withdrawal}\,natural - WW.upstream}{population.local}$ .

The water scarcity status was categorized as

-

-

 using average annual water availability from 1981 to 2010. The 30-year period was used to capture the current hydro climatic characteristics. Fig. 3a represents scarcity for the three water availability types for the Oder basin under average conditions, shown within the Falkenmark matrix (Falkenmark et al. 2007, Kummu et al. 2016) which shows stress and shortage together. Archetypes in the Falkenmark matrix describe the water scarcity status (corresponding to position on the plot) and where both shortage and stress occur, according to which occurs first (Kummu et al. 2016).

None of the SBAs have any shortage as the per capita water availability  has never dropped below 1000 m³ cap⁻¹ yr⁻¹. OdSBA_PO-A is stressed (S) under all three water availability types, and OdSBA_CZ is not stressed (N). OdSBA_GE and OdSBA_PO-B would both be stressed (S)

only if they were restricted to their local runoff (Fig. 3a). After accounting for inflows from upstream (natural discharge), the stress level decreased from 0.25 to 0.01 (N) for $OdSBA_{GE}$ and from 0.35 to 0.01 (N) for $OdSBA_{PO-B}$ (Fig. 3a). This change in stress category means that both these SBAs is dependent on upstream water to avoid stress. We further see that upstream WW increases the stress level relative to natural conditions (to 0.02 for both) (Fig. 3a & c), but the threshold for stress was not crossed. The stress level changed without changing the stress category, such that the category of the dependency was not affected; we have an 'unbroken' rather than 'broken' dependency (definitions in Table 1). In the case of $OdSBA_{PO-A}$, local runoff is not sufficient to meet needs and that upstream water availability and water withdrawals do not influence the scarcity category of this SBA . This SBA is under the same scarcity conditions regardless of upstream influence (Fig. 3a & b), and it is thus categorized as *'No dependency'*, though the intensity of scarcity is still affected by upstream WW. The dependency category of an SBA can then be summarised using three letter codes representing the scarcity category using local runoff, natural discharge and actual discharge respectively: $OdSBA_{GE}$ and $OdSBA_{PO-B}$ are SNN, $OdSBA_{PO-A}$ is SSS, and $OdSBA_{CZ}$ is NNN (Fig. 3a). As illustrated by the above discussion, we can interpret possible scarcity an SBA faces using four different dependency categories as shown below in Fig 4.

A. Scarcity matrix for Oder sub-basin areas (SBAs)

B. Water availability and requirements in Oder sub-basin areas (SBAs)

 3 *Scarcity*

-

-

-

[revised manuscript text omitted]

'No dependency' is observed in 8893% of cases for stress and 9196% of cases for shortage (Table 3).(Table 3).  It is worth noting that scarcity can still be experienced without a dependency – it simply means that current upstream inflows (and WWs) do not influence whether scarcity occurs. ThereFor example, in the case of water stress, 41% of the population living in SBAs under 'no dependency' are under stressed conditions (Table 3). Further, even if an SBA in question is under no dependency category, upstream WW might still intensify the possible scarcity. In this category, there is not currently a problem with relationships with upstream basinsSBAs, but to plan ahead, we need to understand how the situation could evolve, as will beis discussed in Sect. 3.2.4.1.

'ContinuousUnbroken dependency' is observed for both stress and shortage mostly in Africa, Europe,and some parts of Southeast Asia, and North America (Fig. 7aEurope (Fig. 6a, b). ContinuousUnbroken dependency means that maintaining good relationships and assessing water use and potential changes with upstream basins are important. Many to avoid scarcity. A number of SBAs, in which currently no scarcity is observed (Fig. 7),(Fig. 6), are actually suffering fromsubject to upstream dependency. If inflows were to decrease sufficiently due to increased upstream WWs, scarcity could occur, or become persistent rather than occasional.. In these SBAs, this doeshas not yet happenhappened, though upstream WWs may be influencing the frequency and intensity of scarcity, and the level of development (population or use) at which thresholds occurs. Therefore, understanding of how the situation can evolve is needed to know how to manage the relationship with upstream water users.

[Figure]

(a) Sub-basins under different dependency category incase of stress

(b) Sub-basins under different dependency category incase of shortage

No dependency    Continuous dependency    Intervened dependency

NNN  OOO  **SSS**     ONN  SNN  *SOO*     ONO  SNO  **SNS**  **SOS**

[revised manuscript text omitted]

---

## Author Response (AR2)

HESS-2017-537

How downstream sub-basins depend on upstream inflows to avoid scarcity: typology and global analysis of transboundary rivers

**Responses to Editor's comments to the Author:**

**In general I agree with the way in which the authors have addressed the comments by the two reviewers and the editor. The result is that large parts of the text have been modified, and that the argument has been made more straightforward, yet the essence of the original paper has remained. In my view the paper is nearly ready for acceptance. However, I have still the following concerns that need to be addressed:**

**Response:**

We sincerely thank the editor for his valuable comments and feedback.
Below we present the point-by-point responses to the editor's specific concerns.

**Comment 1: I do not find the new names of the concepts of "unbroken dependency" (was "intervened") and "broken dependency" (was "continuous") felicitous, and even counter intuitive. (Isn't a broken dependency a dependency that has been broken, i.e. a dependency that no longer exist? So for me an utterly confusing combination of words). So I urge the authors to find a more straightforward nomenclature. When reading the revised document, I came up with "hidden" and "open" dependency. In fact the manuscript frequently equates unbroken dependency with hidden dependency in the text (6 times, on pages 2, 3, 9, 13 (2x) and 14), apparently because "hidden" clearly conveys its meaning.**

**Response 1:**

Thank you for the suggestion. We agree with the assessment that a downstream sub-basin is still dependent on upstream in the "broken dependency" case, even though they now experience scarcity: the upstream sub basin could alleviate downstream water scarcity by reducing their water use. The term "broken" is indeed inappropriate in that case, and we have adopted the terms "hidden" and "open" throughout the manuscript.

Table 1 has also been edited to emphasis that "open dependency" contrasts with "hidden dependency". The new definition of 'No dependency' reads:

> "Upstream inflows do not influence whether or not a region experiences scarcity, i.e. if a region experiences scarcity or not with only local runoff, additional water from upstream does not change this situation, nor do the upstream water withdrawals. Note that the severity of scarcity may still be affected by upstream inflows and water withdrawals."

The new definition of 'hidden dependency' reads:

> "Scarcity category is altered by upstream inflows but not by upstream water withdrawals, i.e. Local runoff is not enough to meet the local demand but additional water from upstream means the region experiences no scarcity instead of scarcity. Upstream withdrawals are small enough not to change the scarcity status."

The new definition of 'open dependency' reads:

"Scarcity category is altered after accounting for upstream water withdrawals, i.e. while upstream inflows in the hidden dependency allowed the region to avoid scarcity, upstream withdrawals now mean that the SBA does experience scarcity and more intense water management regimes are needed downstream."

**Comment 2: The definitions of dependency must be formalised and unambiguously defined (Table 1 does not provide definitions, merely descriptions). I propose something as follows (using the definitions of discharge runoff as in Table 1; but please check whether my rendering is correct):**

**No dependency: local demand < local runoff OR local demand > actual discharge**
**Dependency: local runoff < local demand < actual discharge**
**Unbroken (hidden) dependency: local demand < actual discharge**
**Broken (open) dependency: local demand < natural discharge**

**Response 2:**

We agree it is useful to add formal definitions in Table 1, to complement visual definitions in Figure 4 in terms of avoidance of scarcity and Figure 5 in terms of water availability thresholds. The correct definitions are:

No dependency: local demand ≤ local runoff OR local demand ≥ natural discharge
Dependency: local runoff < local demand < natural discharge
Hidden dependency: local runoff < local demand ≤ actual discharge
Open dependency: actual discharge < local demand < natural discharge

They have been added to table 1, Page 3 L100.
These definitions are, however, derived from our analysis rather than being pre-determined. We have therefore modified the caption of *Table 1* to say-
"Note: definitions in terms of water availability volumes emerge from our analysis, as described in Section 2 & summarised visually in Figure 4. Our analysis method did not consider the case where actual discharge may be greater than natural discharge." (Page 3, L98)

**Comment 3: Section 2.2.1 describes the method of expressing water availability, but it only deals with the spatial dimension. The temporal dimension is entirely missing. So the reader can only guess what temporal resolution is used. This must be included.**
**Related to this, I would add a sentence on page 7 lines 210-211, namely that these annual values may mask water scarcity during the dry season(s)**

**Response 3:**

Thank you for the suggestion. We have now added the temporal dimension in sec. 2.2.1, Page 6, L175-

"Three types of average annual water availability (for 1981-2010) were calculated in each of these SBAs, corresponding to local water (local runoff), total inflows including upstream areas (natural discharge), and total inflows after upstream WWs (actual discharge) (see detailed definitions in Table 1)."

And also in L180-

"WW for each SBA was calculated separately (referred to as WW.local) by summing up the three water use sectors (industrial, domestic and agriculture) for the year 2010 and aggregating to SBA scale."

We also added the suggested line by the Editor in section 2.2.2, Page 7, L213-214-

"Using annual values may mask water scarcity during the dry season."

**Comment 4: I find Fig 2 confusing and ambiguous with respect to applying the SBA concept: are there three SBASs ("upstream" "middle stream" and "downstream") or are there four SBAs (OdSBAcz, OdSBApoa, OdSBAge, and OdSBApob)? I guess you mean the latter, if so omit the former. If not, clarify and avoid ambiguity.**

**Response 4:**

In this article SBAs (sub-basin areas) were obtained by breaking up the drainage direction map where it flows across country (and shared zone) boundaries, effectively yielding a mesh of river basin and country boundaries. Upstream-downstream relationships between these SBAs were defined by the flow direction dataset as we mentioned in page 5, L166-L169. Afterwards, we categorized the SBAs according to three different types - "upstream", "middle stream" and "downstream". The identified 4 SBAs of the ODER river basin fall under these 3 different types as-

$OdSBA_{CZ}$ -Upstream,
$OdSBA_{PO-A}$ - Middle stream,
$OdSBA_{GE}$ - Middle stream,
$OdSBA_{PO-B}$ - Downstream.

We agree that Figure 2 is not clearly distinguishing the 4 different SBAs currently. In the revised manuscript, we have made the country border more prominent so the definition of the SBAs is more obvious (see below). We also have changed the legend caption from *Sub basin areas (SBAs)* to *Sub basin area (SBA) types*. We hope this will clarify the confusion.

[Figure]

*Revised Figure 2*

**Comment 5: I would combine the last sentence of section 2.2.2 (p. 7 lines 346-347) ("As illustrated …. in Fig.4.") with lines 226-227 on the same page, adding a clarifying remark, for example thus:**

**"The water scarcity status was categorized as No scarcity (N) and Scarcity (S) using average annual water availability from 1981 to 2010. We now can define the earlier defined four different dependency categories in terms of scarcity an SBA can face: NNN, SNN, SNS, and SSS. This is shown in Fig.4."**

**Response 5:**

> We agree with moving the last sentence of section 2.2.2 earlier, but did not want to introduce the three letter names of the categories before explaining how they are obtained. We therefore now added text in page 7, L233-234 as:
>> "Figure 4 defines the four possible different dependency categories in terms of scarcity a SBA can face, as illustrated by the discussion below:"

**Comment 6: Fig 3: in the caption the concept of "water requirements" is used, but this is not defined in the text. Do you simply mean "WW". Please be consistent and parsimonious with concepts! Also Falkenmark's threshold value should have the unit of m3/cap/year.**

**Response 6:**

We agree that "water requirements" was an extra concept, only used for the purpose to visualize water availability a SBA required to avoid stress. To be precise, "water requirements" referred to the water availability required to achieve a water stress value of 0.2 (the threshold used in the paper). We have the following derivation:

stress = WW/avail

avail = WW/stress

avail = WW/0.2

This provided an easy visualization of how water availabilities relate to the stress threshold, but is not a crucial concept otherwise in the paper. To avoid introducing new terms, we have reworded the legend item to read:

"Water availability required to achieve stress=0.2, given local water withdrawals in year 2010"

We hope this will clear the confusion. We have also corrected the unit of the Falkenmark's threshold in the caption. Thank you for pointing this out.

**Comment 7: The last paragraph of section 4.1 (p. 13, lines 400-410) could also refer to the no harm principle (article 7) of the UN Watercourses Convention of 1997.**

**Response 7:**

Thank you for the suggestion. Reference has been added in section 4.1 (Page 14, L 410-412)-

"The UN Watercourses Convention of 1997 also refers to the no harm principle (article 7), which works in tandem with consideration as to whether a given water use is reasonable and equitable (UN 2018)."

**Comment 8: The last two sentences of section 4.2 seem to contradict each other. Please clarify.**

**Response 8:**

Thank you for pointing this out. The sentences have been reworded as:

[revised manuscript text omitted]

**Case study: Oder basin**

Sub basin areas (SBAs)
- Upstream
- Middle stream
- Downstream

Rivers
- River network
- Flow direction at country border

Water availability [$10^9$ m³ yr⁻¹]
- Local runoff
- Natural ('*Local runoff*' + '*Upstream runoff*')
- Actual ('*Natural*' – '*Upstream water use*')

OdSBA$_{PO-A}$

*basin* | *country* | *country sub-code*

**Case study: Oder basin**

Sub basin area (SBA) types
- Upstream
- Middle stream
- Downstream

Rivers
- River network
- Flow direction at country border

Water availability [$10^9$ m³ yr⁻¹]
- Local runoff
- Natural discharge ('*Local runoff*' + '*Upstream runoff*')
- Actual discharge ('*Natural*' – '*Upstream water use*')

OdSBA$_{PO-A}$

*basin* | *country* | *country sub-code*

[revised manuscript text omitted]